# Task-evoked activity quenches neural correlations and variability across cortical areas

Takuya Ito[1,2]*, Scott L. Brincat[3], Markus Siegel[4,5,6], Ravi D. Mill[1], Biyu J. He[7,8], Earl K. Miller[3], Horacio G. Rotstein[1,9,10], Michael W. Cole[1]

**1** Center for Molecular and Behavioral Neuroscience, Rutgers University, Newark, New Jersey, United States of America, **2** Behavioral and Neural Sciences PhD Program, Rutgers University, Newark, New Jersey, United States of America, **3** The Picower Institute for Learning and Memory, Department of Brain and Cognitive Sciences, Massachusetts Institute of Technology, Cambridge, Massachusetts, United States of America, **4** Centre for Integrative Neuroscience, University of Tübingen, Tübingen, Germany, **5** Hertie Institute for Clinical Brain Research, University of Tübingen, Tübingen, Germany, **6** MEG Center, University of Tübingen, Tübingen, Germany, **7** Neuroscience Institute, New York University, New York, New York, United States of America, **8** Departments of Neurology, Neuroscience and Physiology, and Radiology, New York University, New York, New York, United States of America, **9** Federated Department of Biological Sciences, Rutgers University, Newark, New Jersey, United States of America, **10** Institute for Brain and Neuroscience Research, New Jersey Institute of Technology, Newark, New Jersey, United States of America

* taku.ito1@gmail.com

**Data Availability Statement:** All fMRI human data is publicly available through the Human Connectome Project (http://www.humanconnectomeproject.org). All code related to

## Abstract

Many large-scale functional connectivity studies have emphasized the importance of communication through increased inter-region correlations during task states. In contrast, local circuit studies have demonstrated that task states primarily reduce correlations among pairs of neurons, likely enhancing their information coding by suppressing shared spontaneous activity. Here we sought to adjudicate between these conflicting perspectives, assessing whether co-active brain regions during task states tend to increase or decrease their correlations. We found that variability and correlations primarily decrease across a variety of cortical regions in two highly distinct data sets: non-human primate spiking data and human functional magnetic resonance imaging data. Moreover, this observed variability and correlation reduction was accompanied by an overall increase in dimensionality (reflecting less information redundancy) during task states, suggesting that decreased correlations increased information coding capacity. We further found in both spiking and neural mass computational models that task-evoked activity increased the stability around a stable attractor, globally quenching neural variability and correlations. Together, our results provide an integrative mechanistic account that encompasses measures of large-scale neural activity, variability, and correlations during resting and task states.

## Author summary

Statistical estimates of correlated neural activity and variability are widely used to characterize neural systems during different states. However, there is a conceptual gap between

this study can be found at: https://github.com/ito-takuya/corrQuench.

**Funding:** MWC acknowledges support by the US National Institutes of Health under awards R01 AG055556 and R01 MH109520. (www.nih.gov) MS acknowledges support from the European Research Council (ERC StG335880), and the Centre for Integrative Neuroscience (DFG, EXC 307). (https://erc.europa.eu/) EKM acknowledges support from the US National Institutes of Health under award NIMH R37MH087027, the US Office of Naval Research under award ONR MURI N00014-16-1-2832, and the MIT Picower Institute Innovation Fund. The funders had no role in study design, data collection and analysis, decision to publish, or preparation of the manuscript.

**Competing interests:** The authors have declared that no competing interests exist.

the use and interpretation of these measures between the human neuroimaging and non-human primate electrophysiology literature. For example, in the human neuroimaging literature, "functional connectivity" is often used to refer to correlated activity, while in the non-human primate electrophysiology literature, the equivalent term is "noise correlation". In an effort to unify these two perspectives under a single theoretical framework, we provide empirical evidence from human functional magnetic resonance imaging and non-human primate mean-field spike rate data that functional connectivity and noise correlations reveal similar statistical patterns during task states. In short, we found that task states primarily quench neural variability and correlations in both data sets. To provide a theoretically rigorous account capable of explaining this phenomena across both data sets, we use mean-field dynamical systems modeling to demonstrate the deterministic relationship between task-evoked activity, neural variability and correlations. Together, we provide an integrative account, showing that task-evoked activity quenches neural variability and correlations in large-scale neural systems.

## Introduction

Measures of neural correlations and variability are widely used in neuroscience to characterize neural processes. During task states, neural variability has consistently been shown to be reduced during tasks across human functional magnetic resonance imaging (fMRI) [1–3], local neural populations [4–6], and both spiking [5,7] and mean-field rate models [8,9]. Despite this convergence in the neural variability literature, there are disparities in the use and interpretation of neural correlations. In the human fMRI literature, neural correlations are often estimated by measuring the correlation of blood oxygenated level-dependent (BOLD) signals and is commonly referred to as functional connectivity (FC) [10]. In the non-human primate (NHP) spiking literature, neural correlations have been measured by computing the correlation between the spike rate of two or more neurons and is commonly referred to as the spike count correlation (or noise correlation) [11]. Yet despite the use of different terms, the target statistical inference behind these two techniques is consistent: to characterize the interaction among neural units.

In the human fMRI literature, studies have identified large-scale functional brain networks through clustering sets of correlated brain regions using resting-state activity [12–14]. During task states, the FC structure has been demonstrated to dynamically reconfigure [15–17]. Though it has been suggested that correlation increases and decreases respectively facilitate and inhibit inter-region communication [17,18], the mechanistic bases of these FC changes remain unclear.

Studies in the local circuit literature using electrophysiological recordings in animals have characterized the correlation structure between neuron spikes across a range of task demands. These studies have found that the spike count correlation (or noise correlation) between neuron spikes generally decreases during task states, particularly for neurons that are responsive to the task [19–22]. Moreover, these empirical studies have been accompanied by theoretical work, which has suggested that the reduction in noise correlations may enhance information coding by suppressing shared spontaneous activity and reducing neural noise [11,23–26]. Thus, the theoretical framework behind noise correlations may also provide a solid foundation from which to advance understanding of fMRI FC [11,23,25,26].

Here we sought to quantify the relationship between neural correlations (i.e., FC) in large-scale human imaging and (local circuit) animal neurophysiology (spike count correlations). In

particular, it is unclear whether observations at the local circuit level would be consistent with observations made across large cortical areas. To further complicate this issue, we recently found that task activations can inflate task functional connectivity estimates in human fMRI data [27], suggesting some previous neuroimaging studies may have erroneously reported correlation increases due to inaccurate removal of the mean-evoked response. Importantly, the removal of the mean task-evoked response is a standard procedure in the spiking literature, a critical step designed to dissociate signal correlations (task-to-neural associations) from noise correlations (neural-to-neural associations) [11,23,25]. (In the fMRI literature, signal correlations and noise correlations are both statistically and conceptually analogous to task co-activations and functional connectivity, respectively.) Thus, to accurately bridge the FC literature with the spike count correlation literature, it was necessary to analyze the data in a statistically consistent way. This enabled us to adjudicate the differing perspectives in the neural correlation literature while simultaneously confirming and extending previous findings on task-state neural variability reduction.

We report multiple sources of empirical and theoretical evidence demonstrating that task-evoked activity quenches neural correlations and variability across cortical areas. First, we characterize task-evoked neural variability and correlations in empirical data using two highly distinct data sets: multi-site and mean-field NHP spike rates and whole-brain human fMRI. This allowed us to test whether there were consistent large-scale variability and correlation changes during task states independent of data acquisition technique. Moreover, this allowed us to take advantage of the more direct neural recording with NHP electrophysiology along with the more comprehensive coverage of human fMRI (in addition to testing for translation of findings to humans). Next, to provide a mechanistic account capable of explaining our empirical findings, we used both spiking and neural mass models to parsimoniously explain variability and correlation suppression across mean-field cortical areas. This led to theoretical insight using dynamical systems analysis, demonstrating that task-evoked activity strengthens the system's attractor dynamics around a stable fixed point in neural mass models, quenching neural correlations and variability. The combination of simultaneously recorded mean-field spike rate recordings from six cortical sites, whole-brain fMRI obtained from seven different cognitive tasks, and dynamical systems modeling and analysis provide a comprehensive account of task-related correlation and variability changes spanning species and data acquisition techniques.

## Results

We first show empirically that task-evoked activity suppresses neural correlations and variability across large cortical areas in two highly distinct neural data sets: NHP mean-field spiking and human fMRI data (Fig 1). This confirmed previous findings showing quenched neural variability during task states in both NHPs and humans [1,2,4,5], while going beyond those previous studies to report globally quenched inter-area task-state neural correlations. In particular, we focused on neural variability and correlation changes across large cortical areas (mean-field) in our electrophysiology data set (rather than between pairs of neurons) given our focus on large-scale neural interactions, and to facilitate a comparison between different correlation approaches (FC in fMRI data and spike count correlation in electrophysiology data). In addition to spatially downsampling our NHP data to evaluate mean-field spike rates in each cortical area, we also temporally downsampled our NHP data to investigate variability and correlation changes across trials (on the order of hundreds of milliseconds), which appropriately matches the sampling rate of our fMRI data (720ms). Moreover, we limited our inferences to neural interactions between cortical areas to simplify the complexity of analyzing

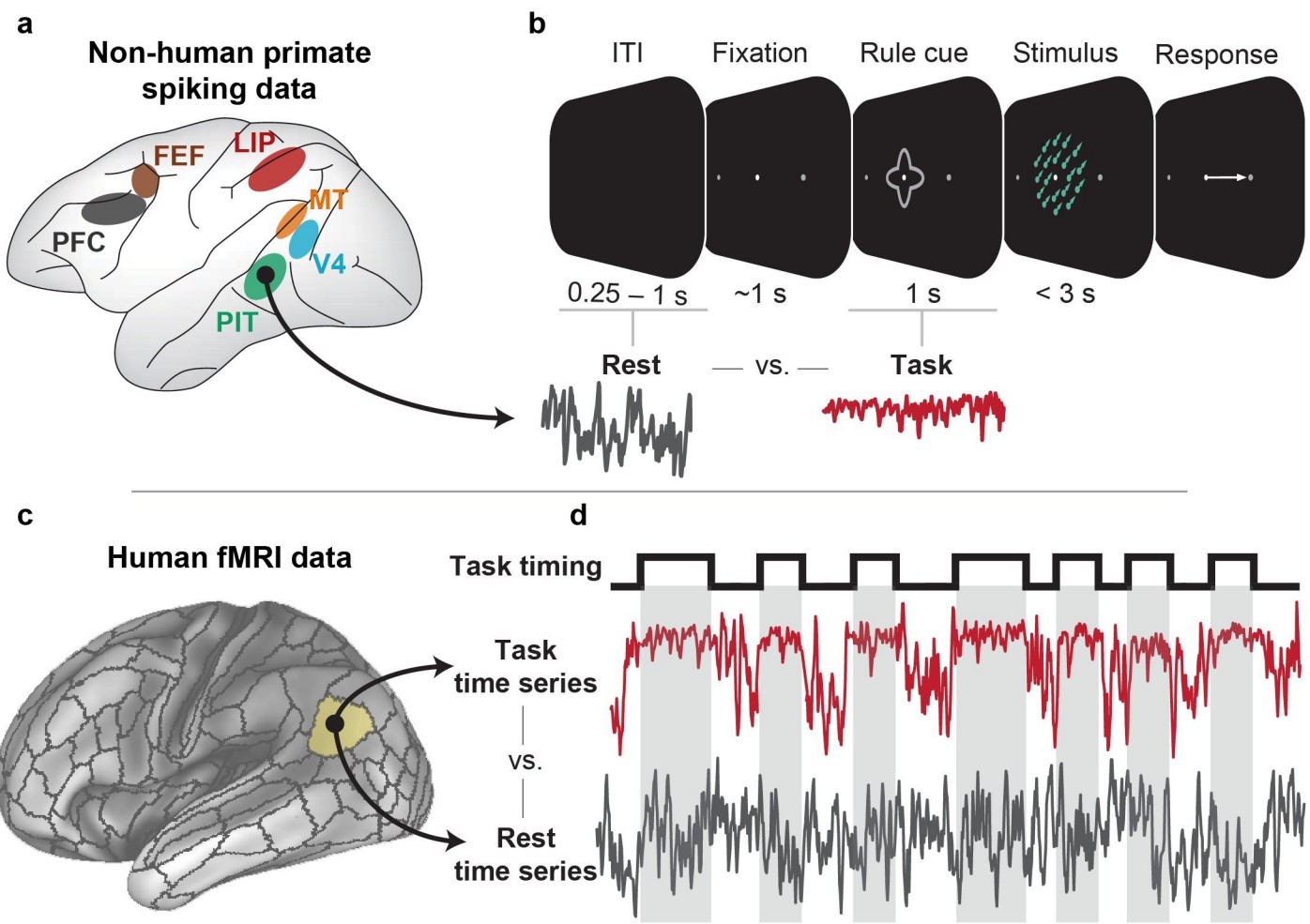

**Fig 1. Testing the hypothesis that task-evoked neural variability and correlations are quenched across cortical areas in NHP spiking and human fMRI data sets.**
We used two highly distinct data sets to test the hypothesis that task-evoked activity globally quenches neural variability and correlations to suppress background
spontaneous activity/noise. This contrasts with the alternate hypothesis, namely that task-evoked activity increases variability and correlation to facilitate inter-region
communication. Importantly, the two data sets were analyzed in a statistically consistent manner, including the removal of the mean task-evoked response to isolate
neural-to-neural interactions. **a,b)** Using mean-field spike rate data collected simultaneously from six different cortical areas [29], we compared the spiking variability
and spike count correlations between task-state (i.e., following task cue onset) and rest-state spiking activity. We defined rest state as the inter-trial interval (ITI)
directly preceding the trial. This was performed by estimating the mean-field spike rate by averaging across multi-units in each cortical area, allowing us to target the
activity of large neural populations. **c,d)** Using human fMRI data obtained from the Human Connectome Project [30], we compared the neural variability and
correlations (i.e., FC) of the BOLD signal during task block intervals to equivalent resting-state intervals. We used seven highly distinct cognitive tasks. Time series and
task timings are illustrative, and do not reflect actual data.

spike count correlations between pairs of local neurons with different receptive fields [21,28].
Following our empirical results, we provide a mechanistic framework using computational
simulations and detailed dynamical systems analyses to explain the quenching of neural vari-
ability and correlations during task-evoked states.

## Task onset reduces neural variability and correlations across spiking populations in NHPs

We estimated the spiking variability and spike count correlations of cortical populations in
NHPs following task cue onset (task periods) and during the inter-trial intervals (ITI) (rest
periods). We found that across trials, global spiking variability and spike count correlations

($r_{sc}$) decreased during task as compared to rest (exploratory subject, variance diff = -3.12, $t_{303}$ = −10.91, p<10e-22, $r_{sc}$ diff = -0.04, $t_{303}$ = −, p<10e-06; Fig 2C and 2D; replication subject, variance diff = -5.37, $t_{807}$ = −13.45, p<10e-35; $r_{sc}$ diff = -0.04; $t_{807}$ = −9.08, p<10e-17; S1 Fig). Variability reductions were also observed using fano factor (rather than variance) at both the mean-field (averaged across neurons; S14 Fig) and for the majority of individual neurons in each cortical area (S12 and S13 Figs). Correlated variability reductions were also observed using spike count covariance (rather than correlations) (Fig 2H). In addition, we demonstrated that variability and correlation decreased within trial (across time points within a trial, after removing the mean task-evoked response), demonstrating that task state quenching also occurs on a moment-to-moment basis, rather than only on a slower trial-to-trial timescale (S2 Fig). We also measured the spiking variability for each cortical area separately, finding that 5/6 cortical areas reduced their spiking variability during task states in the exploratory subject (all areas except for MT, FDR-corrected $p<0.05$). In the replication subject, all cortical areas, including MT, reduced their spiking variability (FDR-corrected $p < 0.0001$). Similarly, we found that during task states, the spike count correlation significantly decreased between a majority of cortical areas (FDR-corrected $p<0.05$; Fig 2D–2G).

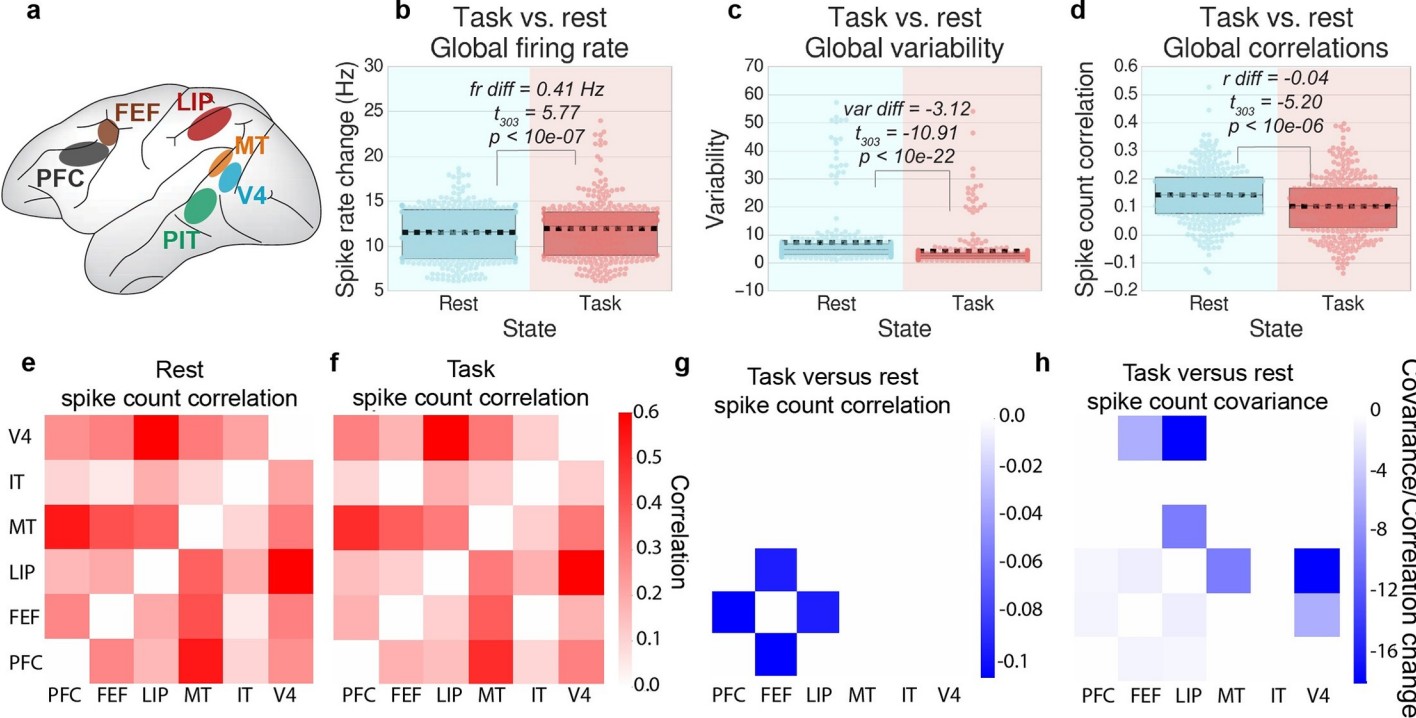

**Fig 2. Neural variability and correlations decrease during task states relative to rest in spiking data.** Results for the replication subject are reported in S1 Fig. **a)** We measured mean-field spike recordings from six different cortical areas during a motion-color categorization task. **b)** We calculated the average spike rate across all recordings during the rest period (ITI) and task period (task cue), across trials. Each data point reflects the firing rate across 25 consecutive trials. **c)** We calculated the cross-trial spiking variance for each region during task and rest states, and then averaged across all regions. Each data point reflects the spiking variance across 25 consecutive trials. **d)** We calculated the average cross-trial neural correlation for task and rest states between all pairs of recorded brain regions. (Spike rates were averaged within each cortical area.) Each data point reflects the correlation across 25 consecutive trials. **e-g)** For each pair of brain regions, we visualize the correlation matrices between each recording site for the averaged rest period, task period, and the differences between task versus rest state spike count correlations. **h)** We also observed no increases in covariance (non-normalized correlation) [31–33]. For panels **e-h**, plots were thresholded and tested for multiple comparisons using an FDR-corrected p<0.05 threshold. Boxplots indicate the interquartile range of the distribution, dotted black line indicates the mean, grey line indicates the median, and the distribution is visualized using a swarm plot. Scatter plot visualizations of **b-d** can be found in S15 Fig.

We did not identify any pairwise correlation and covariance increases in our exploratory NHP (Fig 2G). However, in our replication NHP we found correlation increases between visual and frontal areas (i.e., MT/IT and PFC/FEF) (S1 Fig). When analyzed with covariance (rather than correlation), we found these covariance increases to be weak relative to the observed covariance decreases. (Moreover, the baseline correlation strength between these areas was very low during the ITI period.) Though these correlation increases were only observed in 1 of 2 NHPs, they were generally consistent with our fMRI data (below), which showed that though there were few correlation increases, variability and correlations across cortex were dominated by decreases during task states

To ensure that correlation and variability decreases were associated with increases in the mean activity (rather than just the task period), we estimated the mean spike rate across all regions during the task cue interval and the preceding ITI. Indeed, we found that the mean spike rate during task states was significantly greater than the mean spike rate during rest (exploratory subject, task vs rest firing rate difference = 0.41 Hz, $t_{303}$ = 5.77, p<10e-06, replication subject, rate difference = 0.50 Hz, $t_{807}$ = 3.93, p<10e-04). These findings suggest that task states increase neural activity while quenching spiking variability and spike count correlations across large cortical areas.

Importantly, to accurately dissociate first order statistical effects (mean) from second order effects (variance and covariance/correlation), we removed the cross-trial, mean-evoked response for each task condition. This essential step, which removes the main effect of task, is standard procedure in the spike count (noise) correlation literature [25]. This procedure isolated the underlying spontaneous/background neural activity during task states, which was subsequently used to infer neural interaction through spike count correlation analysis [11]. To ensure consistency between our spiking and fMRI analysis, it was critical that we also carefully removed the mean-evoked response associated with task blocks in our fMRI data (i.e., the main effect of task; see Methods) [27]. To maintain additional consistency between task and rest states in both data sets, we applied the same statistical procedure to our rest data (for both spiking and fMRI data) to control for the possibility that our findings were associated with artifacts related to this procedure (see Methods). (However, we note that the "mean effects" removed as a result from this step during rest periods were negligible.)

## Task-state variability is globally quenched across a wide battery of tasks in human fMRI data

Consistent with the spiking literature, previous work in the fMRI literature has demonstrated that increased activity associated with task-evoked states quenches neural variability [1,2,4]. We extended those findings to evaluate variability quenching across seven additional cognitive tasks in humans using data from the Human Connectome Project (HCP) [34]. We calculated the variability (estimated using time series variance) during task blocks, averaged across tasks and across regions. Consistent with previous reports, we found that the global variability during task blocks was significantly lower than the variability during equivalent periods of resting-state activity (exploratory cohort variance difference = -0.019, $t_{175}$ = −23.89, p<10e-56; replication cohort variance difference = -0.019, $t_{175}$ = −20.72, p<10e-48; Fig 3A). These findings suggest that task states are associated with task-evoked variability reductions.

To better understand how global this phenomenon was, we plotted the change in variability from rest to task for each brain region separately. We found that almost all brain regions significantly reduced their variability from rest to task, suggesting that variability reduction occurs across most brain regions (cortical maps are thresholded using an FDR-corrected threshold of $p < 0.05$; Fig 3). This finding extends the work of a previous study in human

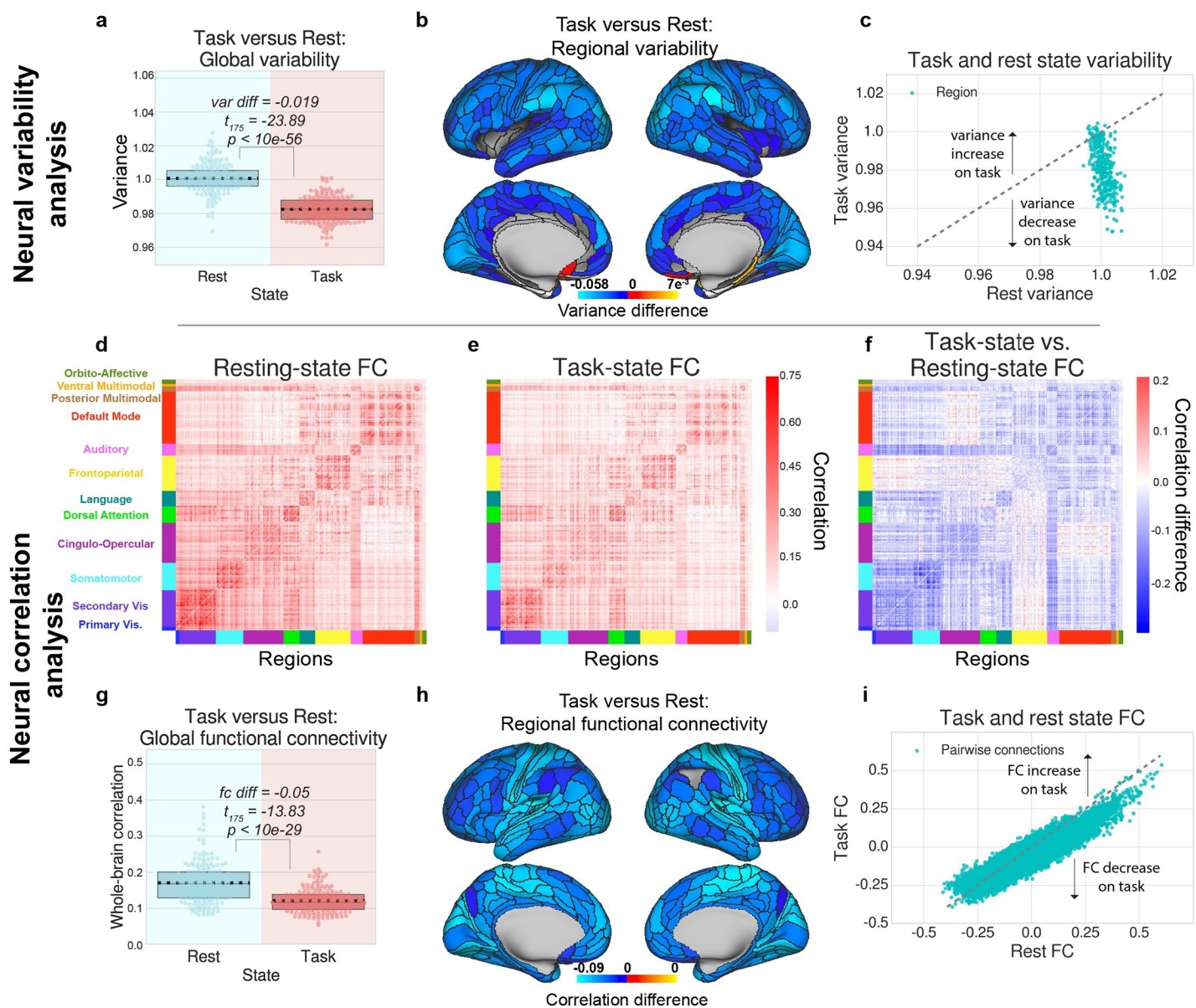

**Fig 3. Variability and correlations decrease during task states in human fMRI data.** Figures for the replication cohort are in S4 Fig. Figures for each task separately are shown in S8 and S9 Figs. **a)** We first compared the global variability during task and rest states, which is averaged across all brain regions, and then **b)** computed the task- versus rest-state variability for each brain region. **c)** Scatter plot depicting the variance of each parcel during task states (y-axis) and rest states (x-axis). Dotted grey line denotes no change between rest and task states. **d)** We next compared the correlation matrices for resting state blocks with (**e**) task state blocks, and (**f**) computed the task- versus rest-state correlation matrix difference. **g)** We found that the average FC between all pairs of brain regions is significantly reduced during task state. **h)** We found that the average correlation for each brain region, decreased for each brain region during task state. **i)** Scatter plot depicting the FC (correlation values) of each pair of parcels during task states (y-axis) and rest states (x-axis). Dotted grey line denotes no change between rest and task states. For panels **b-f,** and **h,** plots were tested for multiple comparisons using an FDR-corrected $p < 0.05$ threshold. Boxplots indicate the interquartile range of the distribution, dotted black line indicates the mean, grey line indicates the median, and the distribution is visualized using a swarm plot.

fMRI data during a finger tapping task [1,8], suggesting that task-induced variability reduction is a general phenomenon consistent across most cortical regions, and across a wide variety of cognitive tasks.

Lastly, we evaluated whether variability quenching occurred during task blocks relative to inter-block intervals (rather than comparing task runs to resting-state runs). Since we z-

normalized each task run with unit variance, we could evaluate the degree to which variability was quenched during task blocks relative to inter-block intervals by computing the average variance during task blocks relative to 1. (Note that z-normalization of the task time series was performed after removing the mean task-evoked response via a task GLM, such that reduced variability was not an artifact of preprocessing/z-normalizing the time series.) Indeed, we found that the variance during task blocks was reduced relative to the inter-block intervals (exploratory cohort variance −1 = -0.019, $t_{175} = −36.58$, p<10e-83; replication cohort variance difference = -0.018, $t_{175} = −33.01$, p<10e-76). Our findings demonstrate that task-evoked periods quench neural variability relative to both resting-state activity and inter-block intervals.

## Task-state FC is globally quenched across a wide battery of tasks in human fMRI data

Despite multiple studies describing task-evoked FC changes [15–17], the precise mechanisms of how FC can change remain unclear. Our current findings illustrate that mean-field spike count correlations decrease during task-evoked states, consistent with previous literature that focused on local circuits [4,11]. Consistent with the spiking literature's perspective on spike count correlations, and the theoretical evidence suggesting that the correlation of ongoing spontaneous activity should be suppressed during task to facilitate information coding [23], we hypothesized that FC would also be globally reduced during task states. To ensure consistency in the statistical analysis across spiking and fMRI data, we removed the mean task-evoked response using a finite impulse response (FIR) model. This approach is statistically equivalent to removing the cross-trial mean response of a task condition, and is a critical step when calculating noise correlations in the spiking literature [11]. This step characterizes the correlation of the background spontaneous neural activity (i.e., background connectivity in fMRI), dissociating task-to-neural interactions (main effect of task) from neural-to-neural interactions (FC) [35].

We first calculated the mean FC across all pairwise correlations across all cortical regions for both task and rest states (Fig 3D–3F). We found that during task states, the global FC was significantly reduced relative to resting-state fMRI (exploratory cohort FC diff = -0.05, $t_{175} = −13.83$, p<10e-29; replication cohort FC diff = -0.046, $t_{175} = −14.00$, p<10e-29; Fig 3G). Recent studies have suggested that the use of correlation provides an ambiguous description of how shared variability (relative to unshared variability) change between brain areas [31,32]. Thus, to generalize these results, we also calculated FC using covariance rather than correlation, finding that covariance also globally decreases (covariance diff = -192.96, $t_{351} = −27.30$, p<10e-88; S5 Fig). Task-evoked global FC was also reduced in each of the 7 HCP tasks separately (all tasks FDR-corrected $p<0.0001$; S9 Fig). To identify exactly how global this phenomenon was, we plotted the average task versus rest FC change for each brain region (Fig 3H and 3I). We found that nearly all cortical regions significantly reduced their correlation with the rest of cortex during task states. To ensure that correlation differences between rest and task states were not associated with in-scanner head motion, we calculated the average number of motion spikes during rest and task scans using a relative root mean squared displacement threshold of 0.25mm [36]. For both the exploratory and replication cohorts, we found no significant differences in the percentage of motion spikes between rest and task states (exploratory set, average task = 0.91% of frames, average rest = 0.81% of frames, $t_{175} = 1.08$, $p = 0.28$; replication set, average task = 0.009% of frames, rest = 0.008% of frames, $t_{175} = 1.53$, $p = 0.12$).

While we primarily observed global decreases in FC, a small portion of connections increased their FC during task states (exploratory cohort, 7.59% of all connections; replication cohort, 9.07% of all connections; FDR-corrected $p<0.05$) (Fig 3I, S4 Fig). However, FC

increases were typically limited to cross-network correlations between networks with different functions, where baseline resting-state FC is already quite low (e.g., cingulo-opercular network with the default mode network, or the frontoparietal network with the visual network) (Fig 3D–3F).

## Task state variability and correlation are quenched independently of removing the mean task-evoked response in fMRI data

The above fMRI results employ the use of FIR modeling to remove the mean task-evoked response to compare task- and rest-state correlations/variability. Here we sought to demonstrate that neural variability and correlations are quenched in fMRI data in the absence of any task regression (e.g., FIR modeling). We used an approach that has been previously used to demonstrate variability quenching following task onset, by measuring the cross-trial variance at each time point [2,4]. We employ the same general approach, measuring the variance and correlation across blocks for each time point within the block. Moreover, to obtain statistically comparable estimates of resting state variability/correlations, we measured the cross-block variance/correlation during sham blocks during resting state by applying the identical task block structure to resting-state fMRI data. Critically, the removal of the mean task-evoked response was excluded from preprocessing for this analysis, and the time series were not z-normalized.

We found that cross-block variance for time points during task state were significantly reduced relative to resting state (var diff = -1009.56, $t_{175}$ = −37.34; p<10e-84; Fig 4A and 4B). We also found consistent results for correlations, finding that the cross-block correlation for time points during task state were significantly reduced relative to resting state (r diff = -0.04, $t_{175}$ = −10.91, p<10e-20; Fig 4C and 4D). These results demonstrate that the quenching of correlations and variability during task states are independent of any potential statistical artifacts that result from removing the mean task-evoked response using FIR task regression.

## Task-evoked activations are negatively correlated with neural variability and correlations in human fMRI data

We showed that task states widely reduced neural variability and correlations. We sought to extend this work to directly demonstrate that decreases in neural variability and correlations are associated with changes in task-evoked activation levels. To provide evidence for this hypothesis, we computed the mean task-evoked activation (averaging across all regions). We found that the global activation was significantly greater than baseline across different task states (exploratory cohort, $t_{175}$ = 6.46, p<10e-9; replication cohort, $t_{175}$ = 12.63, p<10e-25), demonstrating that decreases in neural variability and FC were accompanied by global increases in task-evoked activations.

Previous work has shown that regions that have strong task activations (i.e., the magnitude of the task-induced activation, positive or negative) tend to have greater variability reductions [2]. (Task activation magnitudes reflect the deflection of the BOLD activity relative to baseline, or the inter-block interval.) We sought to replicate this effect in the current data set, while extending those results to demonstrate that more task-active regions also tend to reduce their FC during task states. We first correlated regional task-evoked activation magnitude with task-evoked variability reduction (task variance minus rest variance) across regions at the group-level. We found that regions with greater task-evoked activation magnitudes (averaged across tasks) exhibited greater variability reductions during task states, confirming previous findings in a finger tapping task (exploratory cohort rho = -0.32, p<10e-9; replication cohort rho = -0.49, p<10e-22; Supplementary Fig 3A and 3C) [2]. This negative relationship was also observed in 6/7 of the HCP tasks when analyzed separately (FDR-corrected *p*<0.01; S8 Fig).

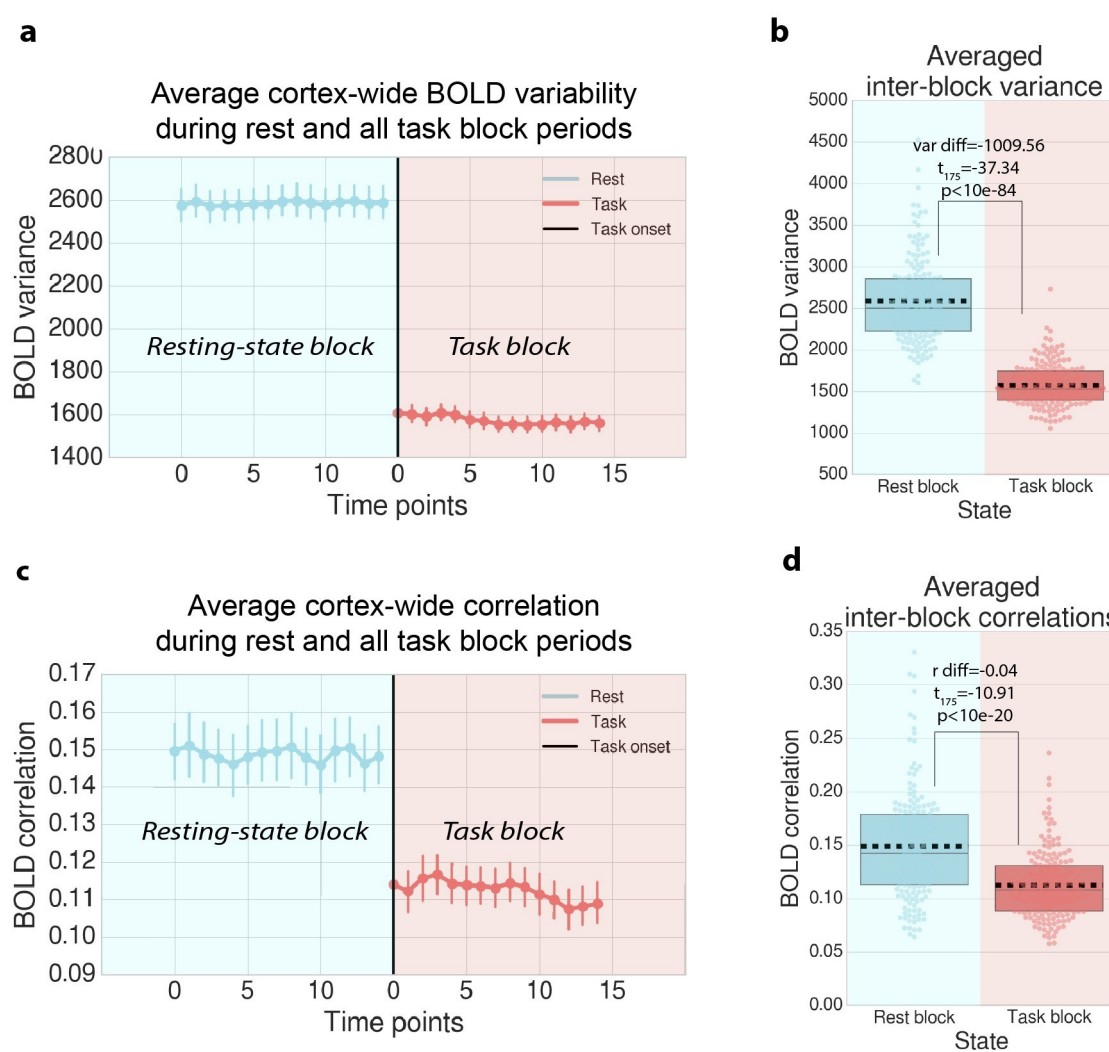

**Fig 4. Task variability/correlations decrease independently of mean task activity removal step in fMRI data.** Instead of computing variance/correlations across time points within task blocks (and removing mean task effects), variance/correlations can be calculated across task blocks (for each time point within a block). This approach isolates ongoing neural activity that is not task-locked, and has been used in both spiking and fMRI data [2,4]. **a)** To isolate ongoing spontaneous activity that is not time-locked to the task, we estimated the variance at each time point across task blocks. The variance at each time point was calculated for each ROI and task condition separately, but then averaged across ROIs and task conditions. Note that to obtain an equivalent variance estimate during resting state, we applied an identical block structure to rest data to accurately compare rest to task state variability. Variability across block time points was averaged across brain regions and task conditions. Error bars denote standard deviation across subjects. **b)** Variance across task block time points was significantly reduced during task blocks relative to identical control blocks during resting-state data. **c)** We performed a similar procedure for task functional connectivity estimates, correlating across blocks for all pairs of brain regions. Correlations across block time points were averaged for all pairs of brain regions and task conditions. **d)** Correlations during task state blocks were significantly reduced relative to identical control blocks during resting state. Boxplots indicate the interquartile range of the distribution, dotted black line indicates the mean, grey line indicates the median, and the distribution is visualized using a swarm plot.

To link regional task activation magnitudes with FC decreases, we tested for a correlation between regional task-evoked activation magnitude and the average FC change during task states for each region. Consistent with our hypothesis, we found that regions with greater task-evoked activation magnitudes (averaged across tasks) reduced their average FC more during task states (exploratory cohort rho = -0.25, p<10e-05; replication cohort rho = -0.20, $p$ = 0.0002; S3 Fig). When tasks were analyzed separately, this negative correlation was

observed in 4/7 of the HCP tasks (FDR-corrected $p<0.05$; S9 Fig). Thus, brain areas with higher levels of task-evoked activation magnitudes (i.e., changes in activity relative to baseline) tend to reduce both their task-evoked variability and FC.

## The information-theoretic relevance of task state reduction of neural correlations

Results from our empirical data converged across imaging modalities and species, illustrating that task states increased mean activity while reducing neural variability and correlations. However, the theoretical implication of a decreased correlated task state remains unclear. Here we sought to better characterize the information-theoretic implication of a global reduction in neural correlations. In particular, consistent with previous large-scale computational models that have predicted increased dimensionality with stimulus-driven activity [37,38], we hypothesized that reductions in neural correlations increase the effective dimensionality across units by suppressing background spontaneous activity/noise. While this increased dimensionality may potentially supports more robust information representations, we acknowledge that a change in neural dimensionality does not necessitate an improvement (or change) in cognitive information representation [23], and that future studies will need to evaluate the relationship between neural dimensionality and cognitive content. Further, we note that an increase in dimensionality is not trivially implied by decreased global correlations. Because we found that regional time series variance also decreases during task states, the neural data dimensionality would increase only if inter-region covariance decreases more than local regional variance (i.e., off-diagonal is reduced more than the diagonal of the variance-covariance matrix).

We measured the dimensionality using the 'participation ratio' [37,39] of the neural activity (for human fMRI and NHP spiking data) during rest and task states (see Methods). Consistent with our hypothesis, we found that task states increased their overall dimensionality relative to rest states (fMRI task versus rest, exploratory cohort difference = 16.13, $t_{175}$ = 19.31, p<10e-44, replication cohort difference = 15.78, $t_{175}$ = 21.66, p<10e-50; NHP task versus rest, exploratory subject difference = 0.13, $t_{303}$ = 5.77, p<10e-07, replication subject difference = 0.26, $t_{807}$ = 13.00, p<10e-34) (Fig 5). We also found that when analyzing each of the 7 HCP tasks separately, dimensionality increased in all 7 tasks relative to resting state (FDR-corrected $p<0.0001$; S10 Fig). The present results suggest that task states were associated with a decrease in neural variability and correlations, reflecting a suppression of shared and private spontaneous activity, which increased the dimensionality of neural activity.

## From neurons to neural masses: Modeling neural dynamics of cortical areas

In the previous sections, we provided empirical evidence that task states reduce mean-field inter-area correlations and variability in spike rate and fMRI data. In this section, we construct a biologically plausible model that provides a parsimonious explanation of correlation and variability reductions in mean field spiking networks and cortical BOLD dynamics.

Neurophysiologically, functional brain areas are composed of local circuits with balanced excitatory and inhibitory neural activity (Fig 6A). In previous work, local circuits have been demonstrated to have clustered excitatory connections [40], leading to slow dynamics and high variability in spiking networks simulated *in silico* [7]. Using this previously established model, we systematically perturbed this balanced network under a distribution of inputs (both excitatory and/or inhibitory inputs) to estimate the excitatory output (i.e., mean-field transfer function) of a cortical population. Though most long-range cortical connections are excitatory, we incorporated excitatory and inhibitory stimulation effects on a local population (Fig 6B).

**a**    *Human fMRI data*          **b**    *NHP spiking data*

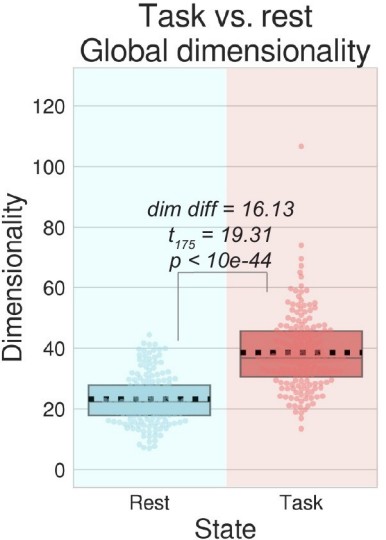
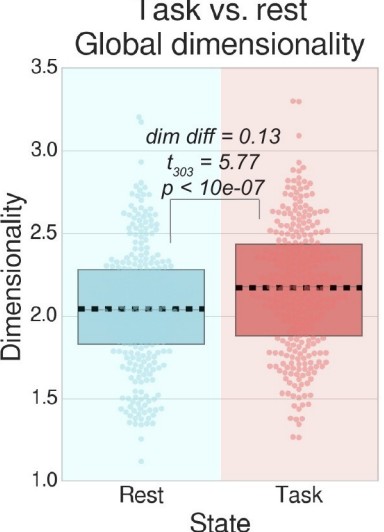

**Fig 5. Dimensionality increases during task periods relative to resting-state activity. a)** For each subject, we calculated the dimensionality using the participation ratio [37,39] during task and rest states and found that during task states, dimensionality significantly increased. **b)** We calculated the dimensionality of spiking activity across trials and found that during task states, dimensionality significantly increased. These findings provide a potential information-theoretic interpretation of neural correlation and variability reduction during task states. Boxplots indicate the interquartile range of the distribution, dotted black line indicates the mean, grey line indicates the median, and the distribution is visualized using a swarm plot.

This is because long-range excitatory afferents may target local inhibitory neurons, producing a net inhibitory effect. Under the presence of inputs, we found that the population transfer function approximated a sigmoid activation function (Fig 6B). We note that the upper bound on the sigmoid transfer function (Fig 6D) is likely due to inhibitory feedback on excitatory activity rather than the true saturating spiking regime in neurons. This is because excitatory neurons in a local population typically do not reach a saturating spiking regime even for strong visual stimuli [41], and instead reach an upper bound due to strong inhibitory stabilization preventing runaway excitation [5]. Importantly, simplifying the mean-field transfer function of a cortical area allowed us to focus our modeling efforts on simplified networks across large cortical areas [42].

In this balanced spiking network, any evoked stimulation, excitatory or inhibitory, would result in reduced variability (Fig 6C). Specifically, the magnitude of stimulation was negatively correlated with spiking variability in the balanced spiking model (rho = -0.92; p<0.0001). While previous studies have suggested that the mean and variance of the spike rate may be independent of each other, those studies focused on mean-matching the spike rate of individual neurons within the same local population [4,7]. However, in this study, we focus exclusively on the mean-field level rather than individual neurons. We found a highly negative association between mean and variance under experimental perturbation, suggesting that at the mean-field level, mean and variance cannot be mechanistically dissociated. Based on these considerations, we hypothesized that during periods in which global neural activity levels are elevated, such as task states, both neural variability and correlations would be globally quenched.

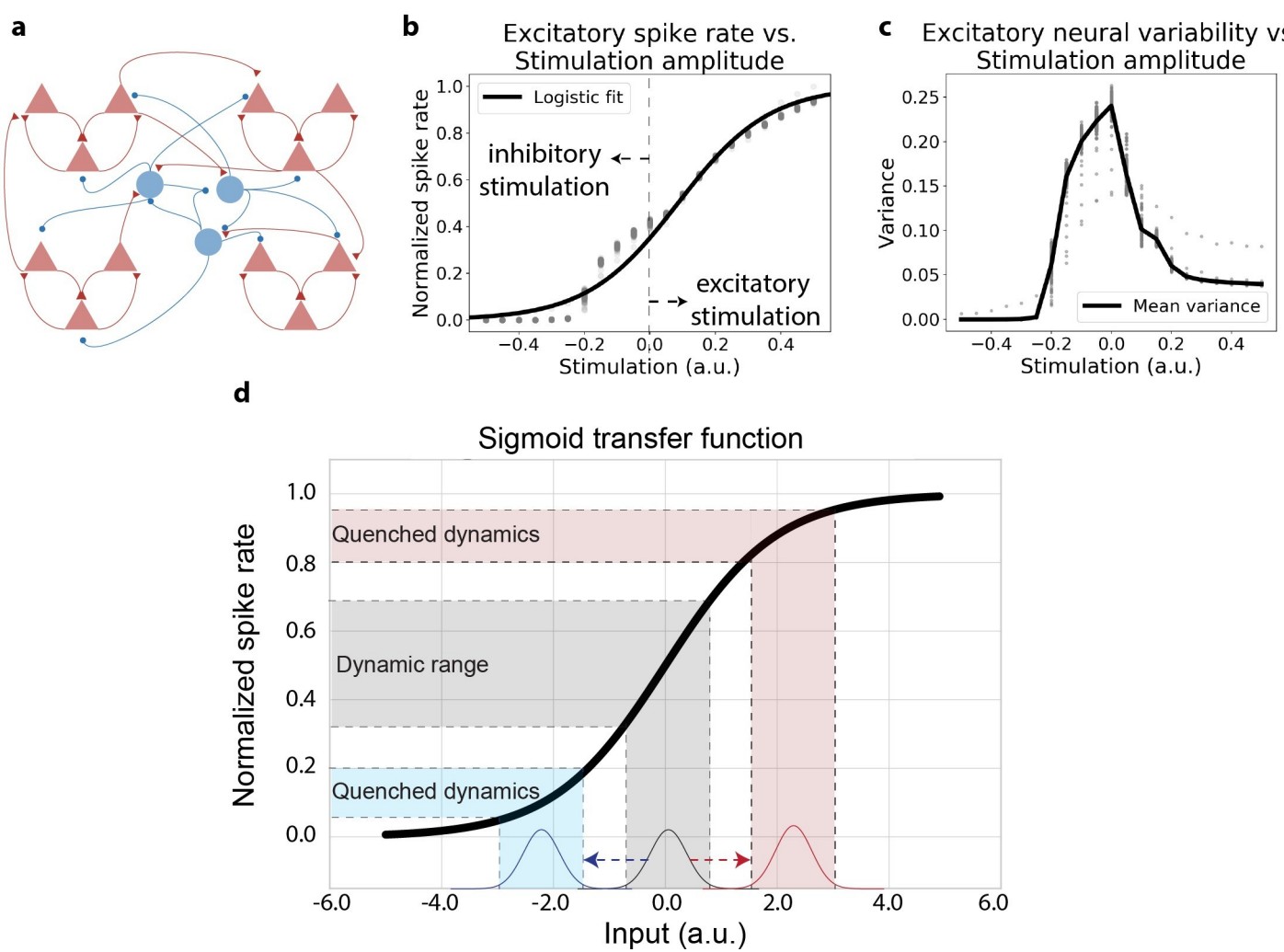

**Fig 6. Inferring the mean-field transfer function of a neural population with a balanced spiking model with clustered excitatory connectivity. a)** Schematic illustration of the balanced spiking model with clustered excitatory connections. Network architecture and parameters are identical to those reported in [7]. Red triangles indicate excitatory cells, blue circles indicate inhibitory cells. **b)** The population spike rate (excitatory cells only) subject to inhibitory regulation. We systematically stimulated a subset of the neural population and measured the corresponding mean excitatory spike rate. Spike rates were normalized between 0 and 1. Excitatory stimulation was implemented by stimulating 400 excitatory neurons, and inhibitory stimulation was implemented by stimulating 400 inhibitory neurons. Spiking statistics were calculated across 30 trials, with each point in the scatter plot indicating a different 50ms time bin. **c)** Population neural variability (excitatory cells only), as a function of input stimulation. **d)** Based on panel **b**, we approximated the mean field neural transfer function as a sigmoid. A sigmoid transfer function produces optimal input-output dynamics for a narrow range of inputs (gray). The same input distribution mean shifted by some excitatory/inhibitory stimulation produces a quenched dynamic range.

## Neural variability is quenched during task-evoked states in a neural mass model

Here we rigorously ground the intuition that task-evoked activity reduces output variability using neural mass modeling and dynamical systems theory. A recent study provided evidence that an evoked stimulus drives neural populations in sensory cortex around a stable fixed point attractor [5]. We first extended these findings using a simplified neural mass model, which allows for a comprehensive dynamical systems analysis that is mathematically difficult in higher dimensions. Additionally, this enabled a simpler theoretical approach to investigating changes in neural dynamics that are generalizable across mean-field neural cortical areas (i.e., populations with sigmoidal transfer functions).

We first characterized the relationship between task-evoked and spontaneous activity in a large neural population using a single neural mass unit. We simulated the neural population's dynamics across a range of fixed input strengths (Fig 7A), finding a nonlinear relationship between stimulus strength and the observed variability of the neural population (Fig 7C). We found that variability was highest when there was no stimulation, while variability decreased for any type of evoked stimulation (e.g., negative or positive input amplitudes). Despite the model's simplicity, these findings are consistent with our (and others') empirical and model results demonstrating that task states quench time series variability in both human and animal data [1,4,5]. We also generalized the findings from our minimal (single region) model to large-scale firing rate models (with 300 regions), where we found variability decreases during task-evoked states in both network models with random structural connections and clustered structural connections (S11 Fig). We demonstrated this for network models with excitatory connections only, as well as networks with both excitatory and inhibitory connections. However, due to the large number of possible network models when scaling to n-dimensions, we constrained

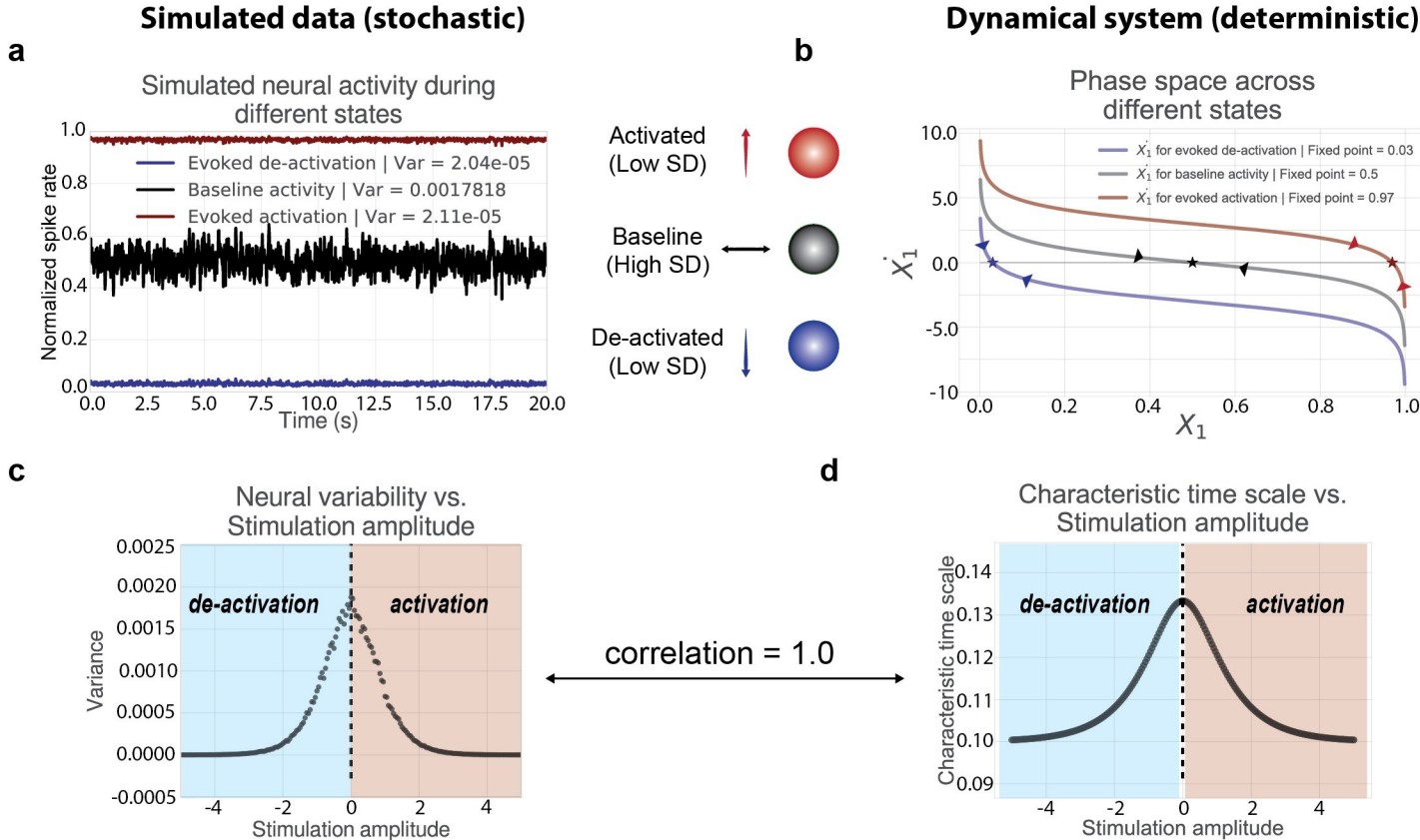

**Fig 7. Task-evoked activity induces changes in neural variability and the underlying attractor dynamics.** Our minimal modeling approach directly links descriptive statistics (e.g., time series variability) with rigorous dynamical systems analysis (e.g., attractor dynamics). **a)** During different evoked states (i.e., fixed inputs), there is a reduction in the observed time series variability (measured by variance across time). This is directly related to how input-output responses change due to the changing slope in the sigmoid transfer function. **b)** We visualized the phase space for each of the neural populations according to state by plotting the derivative of $X_1$ denoted by $\dot{X}_1$. For each state, we estimated the fixed point attractor (plotted as a star), denoting the level of mean activity the system is drawn to given some fixed input (or absence thereof). Arrows denote the direction/vector toward each fixed point, which specify the characteristic time scale (i.e., the speed) the system approaches the fixed point. **c)** We ran simulations across a range of stimulation amplitudes, calculating the variance across time at each amplitude. **d)** We characterized the shifting attractor dynamics for each stimulus by computing the characteristic time scale at the fixed point for each stimulation amplitude. The characteristic time scale across all fixed points are nearly perfectly correlated with the neural variability of the simulated time series across all fixed inputs (rank correlation = 0.9996).

our analyses to only four network architectures, leaving a more complete analysis to future studies.

We sought to leverage the model's simplicity to characterize dynamical systems properties governing the observed neural variability. This would provide rigorous evidence that shifting the underlying attractor dynamics alters the observed neural signals. We first performed a state space analysis in one dimension to identify the stable fixed point attractor (i.e., the equilibrium level of activity the system is drawn to during a particular state) for the intrinsic and evoked states (Fig 7B). The state space view enabled visualization of the system's full dynamics across different evoked states (Fig 7B). For example, dynamics around the fixed point attractor in the intrinsic baseline (rest) state appeared to approach equilibrium slowly. This can be identified by observing the angle where the curve intersects 0 on the y-axis (i.e., when $\dot{x} = 0$; Fig 7B). The angle of this curve corresponds to the *characteristic time scale*, a dynamical property characterizing the speed with which the system approaches the attractor (a higher value reflects slower dynamics; see Methods) [43].

To quantify this more rigorously, we performed a linear stability analysis around the fixed point attractor of the system across the same range of stimulation amplitudes. For each input, we analytically calculated the characteristic time scale at each fixed point. Again, we found a nonlinear relationship between the amplitude of the stimulus and the characteristic time scale of the neural population (Fig 7D), and found that the characteristic time scale explained nearly 100% of the variance (rho = 0.9996) of the simulated stimulus-evoked variability (Fig 7C). These results demonstrate that changes in observed neural variability can be directly attributed to changes in the underlying attractor dynamics.

To ensure that the model explanation would generalize to data obtained on a slower time scale (e.g., fMRI BOLD data), we transformed the simulated neural activity into fMRI BOLD activity using the Balloon-Windkessel model [44]. The Balloon-Windkessel is a nonlinear transformation of neural activity to model the BOLD signal that takes into account the normalized blood volume, blood inflow, resting oxygen extraction fraction, and the normalized deoxyhemoglobin content. Consistent with previous accounts [2], we found that the characteristic time scale around the fixed point attractor was still strongly correlated with BOLD variability (rho = 0.97; $p < 0.0001$; S6 Fig).

## Neural correlations are quenched during task states in a network model

We generalized the dynamical systems analysis in one dimension to two dimensions, allowing us to focus on correlations across cortical areas. We show illustrations of the state space for intrinsic and task-evoked states (Fig 8B and 8D), as well as the corresponding time series (Fig 8A and 8C) for our model. Induced negative activity produced qualitatively similar results to the activated state (Fig 8D) due to subthreshold levels of activity rather than saturating levels of activity.

The state space analysis (Fig 8B and 8D) allowed us to track the simultaneous evolution of the two neural masses, providing a geometric interpretation of the system. We observed qualitatively that shifts in the attractor dynamics (i.e., changes to the flow field) due to stimulation were directly associated with changes to the correlation between the two neural masses. Specifically, we observed that intrinsic state dynamics supported slower, elongated trajectories along a diagonal axis, consistent with correlated neural activity between the two masses (Fig 8B). This was due to a slower characteristic timescale near the fixed point attractor, which corresponds mathematically to eigenvalues with smaller magnitudes. In contrast, during evoked states, the system approached the fixed point attractor at a faster speed, quenching trajectories in state space that supported correlated variability (Fig 8D). Thus, the visualization of the state

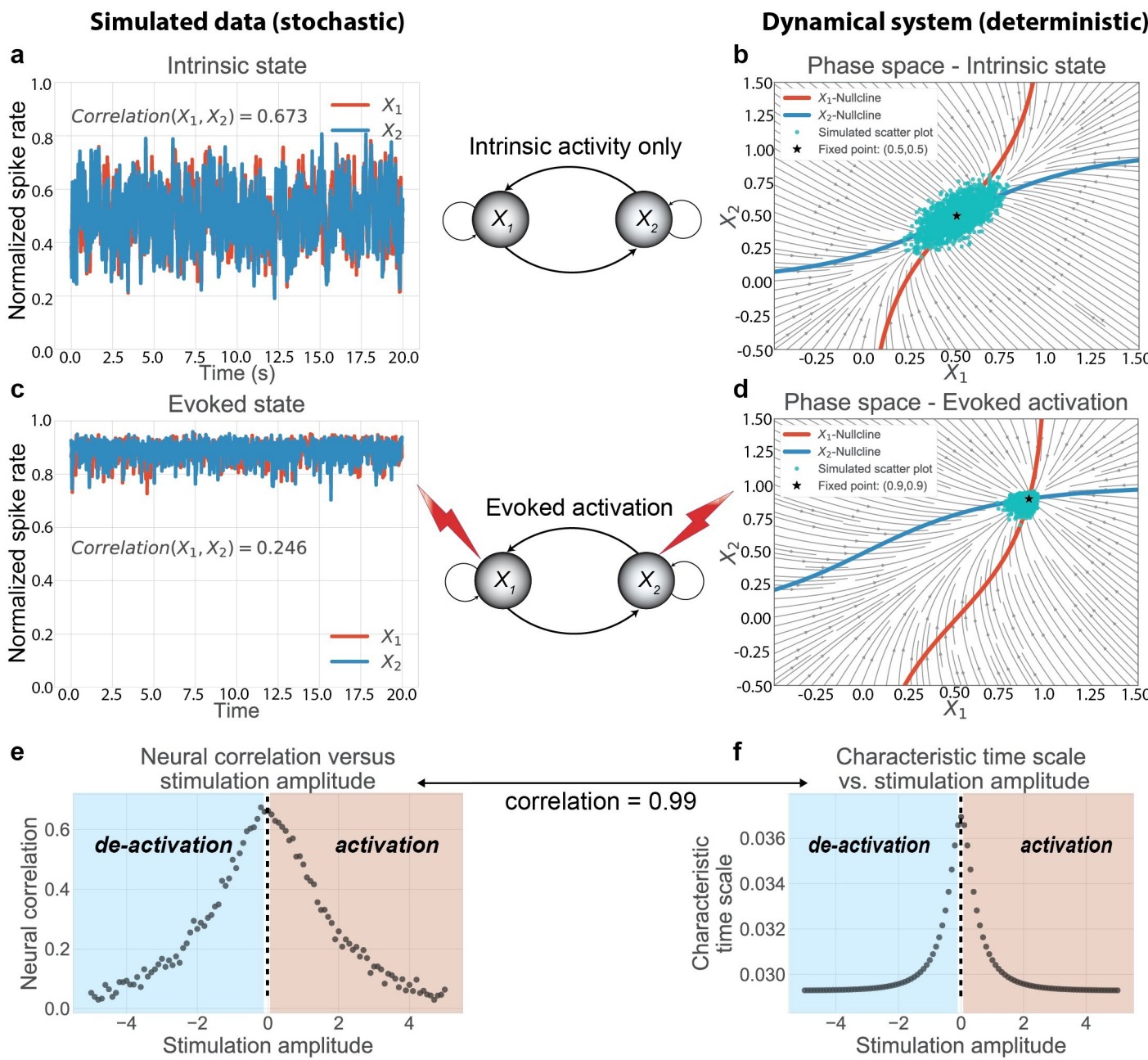

**Fig 8. Task-evoked activity quenches neural correlations by altering the underlying attractor dynamics.** We used a two unit network model, the minimal model necessary to study dynamic changes in neural correlations. **a)** At baseline, we observed slow, high amplitude fluctuations and high neural correlations. **b)** To characterize the underlying attractor dynamics, we visualized the two-dimensional state space, visualizing the flow field and the nullclines (blue and red curves, where the rate of change is 0) for each unit. The intersection of the two nullclines denote the fixed point attractor. We overlaid the simulated scatter plot (cyan dots) to illustrate the correspondence between the attractor dynamics and simulation. **c)** We injected a fixed input stimulation, shifting the network to an 'evoked' state, which caused a decrease in neural variability and correlation. **d)** The external input transiently moved the fixed point, altering the attractor dynamics and the corresponding scatter plot. **e)** We systematically injected a range of fixed inputs into the network. We found that neural correlations were optimal with no external stimulation, and decreased with any external stimulation. **f)** Across stimulation strengths, we found that the generalized characteristic time scale (see Methods) near the fixed point explained 98% of the neural correlation variance, providing a direct association between the network's attractor dynamics and observed neural correlations.

space demonstrated that changes in neural correlations were associated with changes to the flow field around the fixed point attractor.

To more carefully test the relationship between state-dependent neural correlations, we simulated our network model across a range of fixed input amplitudes. Despite no changes to the network's connectivity structure, we found that neural correlations systematically changed (decreased) as a function of evoked stimulation (Fig 8E). Further, using dynamical systems analysis, we found that a generalization of the characteristic time scale in higher dimensions accounted for changes in neural correlations (rho = 0.99; p<0.0001; Fig 8F). In other words, we analytically determined that evoked stimulation shifted the attractor dynamics, changing the neural correlations in a network model with fixed synaptic connections. We found consistent results after transforming the neural activity to fMRI BOLD activity using the Balloon-Windkessel model [44], finding that changes to the characteristic time scale accounted for changes in BOLD correlations (i.e., FC) (rho = 0.97; *p*<0.0001; S7 Fig). These results were reproduced using mutual information (rho = 0.94; p<0.0001), a nonlinear measure of statistical dependence [45], and non-parametric rank correlation (rho = 0.99; p<0.0001). This suggests that the quenching of shared variance encompasses both parametric and non-parametric linear and nonlinear measures of statistical dependencies.

To ensure that our findings in simplified two node networks would scale to large-scale network models, we simulated large-scale firing rate models (with 300 regions). We found correlation decreases during task-evoked states in both network models with random structural connections and clustered structural connections (S11 Fig), suggesting that the mechanisms we identified in these minimal models likely scale to the larger networks. We demonstrated this for network models with excitatory connections only, as well as networks with both E and I connections.

## Discussion

The present results suggest that task-evoked neural activity globally quenches neural time series variability and correlations. We showed this in NHP spiking and human fMRI data, illustrating the generality of the phenomena. This supports the hypothesis that during task states, decreases in neural variability and correlations suppress ongoing spontaneous activity, better supporting information coding [23]. We subsequently provided a dynamical systems model to demonstrate that evoked activity strengthened the system's fixed point attractor, quenching neural variability and correlations. This provided a mechanistic framework to interpret the empirical results. Importantly, the use of a sigmoid transfer function to model mean-field cortical dynamics revealed a simple interpretation underlying neural variability and correlation suppression widely applicable to many types of neural data. During task states, the effective slope of the neural transfer function decreases, reducing the dynamic range of input-output responses. This results in reduced overall output variability, as well as reduced shared variability (e.g., correlations) from connected neural populations. The collective empirical and theoretical results provide strong evidence that observed neural variability and correlations are state-dependent, and these changes emerge from the activity dynamics governed by the transfer functions of large neural masses.

The relationship between neural correlations and neural communication (or FC) is complex. For example, it appears that a decrease in the neural correlation between a pair of brain regions does not simply imply a reduction in communication. In the spiking literature, this interpretation is attributed to a reduction of shared spontaneous activity (or, neural noise). This is because the cross-trial mean evoked response (i.e., "the signal" associated with the task or stimulus) is removed prior to calculating the correlation, leaving only "neural noise" or spontaneous, moment-to-moment activity [11]. Notably, this "neural noise" can still be important, since some portion of it drives trial-by-trial variability in cognition and behavior. In the

fMRI literature, this is equivalent to regressing out the cross-trial mean task-evoked activity associated with the task/stimulus prior to calculating FC [27]. (This type of FC is also referred to as "background connectivity" [35].) The primary reason for this is to target neural-to-neural correlations, rather than task-to-neural associations.

Our theoretical and empirical results clarify the interpretation of correlation changes from rest to task states in large-scale neural systems. Though empirical studies in large-scale functional networks using fMRI have reported FC increases during task states [46], we recently found that task-evoked activity inappropriately inflates FC estimates if the mean-evoked activity response is not properly accounted for [27]. Indeed, when properly accounting for the mean-evoked response, we found that FC changes from rest to task states were dominated by FC decreases (see Fig 3I). The correction of the mean-evoked response in our paradigm brought the empirical results in line with our modeling results, suggesting a counterintuitive interpretation of FC changes during tasks: task co-activation in the presence of neural correlation quenching is consistent with task-related signal communication with background noise suppression. This can be understood from an information-theoretic perspective: during task communication, ongoing spontaneous activity will be suppressed (i.e., neural variability and correlations), increasing the fidelity of the task signal (mean task-evoked response). Our results were consistent using both correlation and covariance measures, suggesting that these decreases were due to reductions in shared variance rather than changes in unshared variance [3,31–33]. Furthermore, the present results do not rely on regressing out the task; correlation and variability quenching were also observed independent of this preprocessing step (see Fig 4). This was achieved by isolating cross-trial variance, which is similar to computing FC with a beta series correlation [47].

Though we largely focused on FC decreases during task states in both data sets, we identified a small number of correlations that increased during task state. Most of these correlation increases were primarily between regions belonging to different functional networks, such as frontal and visual areas, which is consistent with previous literature [15–17]. Some correlation increases have also been reported in the NHP spiking literature, where spike count correlations between units with similar task receptive fields tend to decrease, while spike count correlations between units with dissimilar task receptive fields increase [21]. This appears to be conceptually consistent with the present findings, where we focus on mean-field correlation changes between functionally distinct cortical areas (rather than between pairs of individual neurons). Specifically, we found that regions in the same network tend to decrease their correlations, while regions across functionally distinct areas/networks marginally increase their correlations. In our fMRI data, we found marginal task-state correlation increases between different networks, such as the frontoparietal and visual networks (Fig 3F). Correlation increases were also observed in one of the two NHPs between frontal (PFC and FEF) and visual (MT and IT) areas (S1 Fig). (However, we note we did not observe any correlation increases in our exploratory NHP.) While these correlation increases appear to be statistically reliable, the spontaneous (resting-state) correlations between these areas were very low and the correlation increases marginal. Thus, it will be important for future investigations to directly evaluate the functional relevance of these marginal correlation increases.

We propose that a sigmoid transfer function is an effective model of the activation dynamics of large cortical areas. This was based on causally manipulating a locally balanced E-I circuit with clustered excitatory connectivity [7]. The sigmoid transfer function provides a simplification of mean-field neural dynamics that captures excitatory dynamics subject to inhibitory regulation, where feedback inhibition is implicitly modeled by the saturation of the sigmoid function. (We note that the saturation of the sigmoid function does not represent the saturating spiking regime, since spiking saturation does not typically occur *in vivo* [41]). Moreover,

the implementation of the sigmoid transfer function is consistent with prior computational studies demonstrating that resting-state activity corresponds to dynamic regimes with large amplitude, slow fluctuations [48]. In contrast, during task-evoked states, the output dynamics of the sigmoid transfer function are reduced, which correspond to evoked states (e.g., cortical "Up" or asynchronous states) that exhibit quenched variability [49,50]. The quenching of output variability can be explained by different biological mechanisms, such as clustered excitatory connectivity in local circuits, tightening of E-I balance due to inhibitory feedback, neural adaptation, and/or irregular synaptic vesicle release [5,7,9,26,51,52]. The manifestation of these biological mechanisms can be summarized at the mean-field by the reduction of the response variability due to the decreased slope in the sigmoid transfer function during highly active or inactive states. Though other detailed spiking models have offered biophysical mechanisms for inter-area spiking correlations [53], we focused here on simplified dynamical systems explanations for correlation and variability changes at the mean-field. It will be important for future work to directly investigate how lower-level biophysical mechanisms map onto model descriptions at the mean-field.

The present results may appear to contradict some reports that task engagement increases (rather than decreases) overall neural communication. Yet there are several key differences between those previous findings and the present results. First, many of the previous results focused on communication through coherence, which often involves frequency-specific coupling of neural signals [54]. This involves the phase-alignment of neural activity on faster time-scales, which relates only indirectly to the slower correlation measures of spike rate activity and metabolic demand focused on here [55,56]. A second key difference between the present and most previous results is our emphasis on the absolute amount of correlation change from rest to task, rather than changes in network organization. Previous studies have also acknowledged that global signal regression, a common fMRI preprocessing step, shifts the baseline of rest- and task-state correlations and artificially induces negative correlations [57,58]. This preprocessing step confounds the comparison of the magnitude of correlation changes during independent rest- and task-state fMRI. In the present study, we rigorously preprocessed our fMRI data while ensuring to not remove the global signal. Along with the recent finding that incorrect removal of the mean-evoked response can inflate FC estimates, we suggest that rest to task correlation increases in previous fMRI studies should be interpreted with care.

Despite converging results, there are several key differences in our two empirical data sets. First, the time scale of fMRI BOLD activity is much slower than the NHP spiking activity. However, these differences were mitigated by measuring spiking variability across trials, which is comparable to the time scale of fMRI's sampling rate (in the hundreds of milliseconds). (We note, however, that downsampling spike data does not make it statistically equivalent to fMRI data.) In addition, our computational model results demonstrated that reductions in neural variability and correlations were preserved after nonlinearly transforming the spike rate signal to the fMRI BOLD signal with the Balloon-Windkessel model [44], suggesting that the observed signal changes are likely due to the BOLD signal changes rather than MRI artifacts. Despite the computational model demonstrating that statistical properties of BOLD dynamics can be directly caused by spiking dynamics (via the hemodynamic transformation), it is difficult to rule out other possible *in vivo* explanations in the present study. Another key difference in the data sets is the lack of a true resting-state data set in our NHP data. However, to better compare these two data sets, we demonstrated in the human fMRI data set that the task block periods showed reduced variability relative to inter-block intervals, which is a more analogous comparison to the NHP data set. Despite replication of results across species and task paradigms, our conclusions are based on independently obtained data sets from two species and across task designs. Thus, it will be important for future work to more thoroughly investigate

the differences in variability and correlation quenching using experimental designs that simultaneously record both BOLD signal and spiking activity across multiple cortical areas.

In conclusion, we propose a mechanistic framework for interpreting changes in neural variability and correlations by investigating the effects of task-state activity on the underlying neural attractor dynamics. Using empirical data analysis across two highly distinct neural data sets and theoretical modeling, we demonstrated convergent evidence suggesting that task states quench neural variability and correlations due to strengthening neural attractor dynamics across large-scale neural systems. Our work extends previous research establishing similar attractor mechanisms in sensory cortex [5] to characterize the role of attractor dynamics across large-scale cortical areas. We expect these findings to spur new investigations to better understand how we can interpret neural variability and correlations during task states, providing a deeper understanding of dynamic processes in the brain.

## Methods

### Spiking data: Data collection

The behavioral paradigm for each monkey was a motion-color categorization task (Fig 1B). Experimental methods for electrophysiology data collected for NHP was previously reported in [29] and [59]. Data was collected *in vivo* from two (one female) behaving adult rhesus macaques (*Macaca mulatta*) across 55 sessions. Data from six distinct cortical regions were recorded simultaneously from acutely inserted electrodes. Cortical regions included: MT, V4, PIT, LIP, FEF, and LPFC (Fig 1A). Spikes from each region were sorted offline into isolated neurons. However, given our interest in inter-region neural correlations across large scale neural systems, we pooled spikes from each functional area into a single spike rate time series. For each trial, spikes were sorted for a 5s period, beginning 2.5s prior to stimulus onset, and until 3.5s after stimulus onset. Further details regarding electrophysiological data collection can be found here:

http://www.sciencemag.org/content/348/6241/1352/suppl/DC1 and here:
http://www.pnas.org/content/pnas/suppl/2018/07/09/1717075115.DCSupplemental/pnas.1717075115.sapp.pdf

All statistical analyses in the main article (detailed below) were performed on a single monkey. Independent replication was performed on the second monkey, and is reported in S1 Fig.

### Spiking data: Task versus rest variability analysis

Neural variability analysis was analyzed using an analogous approach to both the computational model and fMRI data. However, since we had no true 'resting-state' activity for the monkey data set, we used the inter-trial interval (ITI; 0.5s - 1s variable duration, see Fig 1B) as "resting-state activity". We used the 0.5s - 1s interval immediately preceding the trial's fixation period to avoid any reward/feedback signals from the previous trial. (Reward/feedback from the previous trial was provided more than 1.5s prior to the fixation period.) Spike counts were calculated by taking a 50ms sliding window with 10ms increments, consistent with previous studies [4]. The mean-evoked response across all trials for a given task rule (e.g., motion rule versus color rule) was calculated and removed from each trial, as is common in the spike count literature [11] and the fMRI literature [27]. (Statistically this is equivalent to performing the task activation regression in the fMRI data, described below.) The mean task-evoked response of the ITI period associated with each task condition was also removed. This was to control for any artifacts that might be induced due to removal of the mean-evoked response. Trials with less than 500ms (or 50 time points) worth of spiking data for either the ITI and/or

task cue presentation were excluded. This was done to reduce variability of the estimated spike count correlations, since correlations with few observations are highly variable.

We computed the variance across 25 consecutive trials using the spike rate from each cortical recording during either the ITI or the task cue period. This was repeated for all trials for each subject. We used across-trial variance to calculate variability rather than Fano factor [4]. This choice was due to the insight from our model illustrating that the mean-evoked activity and the corresponding variance interact in a nonlinear manner, and that the Fano factor is computed as the variance over the mean. Cross-trial variance was computed as

$$Var = \sum_{trial=x}^{x+n} \frac{(r_{trial} - \bar{r})^2}{n-1} \tag{1}$$

Where $n = 25$ trials, $r_{trial}$ reflected the spike rate of each trial, and $\bar{r}$ is the cross-trial mean firing rate for the task condition (i.e., either the cross-trial mean firing rate during the color or motion task cue period).

The statistical difference in task versus rest neural variability was computed by using a two-way, paired t-test across all bins of 25 consecutive trials. The global neural variability change was computed by averaging the variance across all recording areas for each bin. Statistics for the regional neural variability change were corrected for multiple comparisons using an FDR-corrected threshold of $p < 0.05$.

In addition, we computed the variance during the ITI and task cue period within trial (across time points) (S2 Fig). This analysis demonstrated that moment-to-moment variability (rather than trial-to-trial variability) was also quenched from rest to task periods, suggesting that variability quenching also occurs at faster timescales. The statistical difference in task versus rest neural variability was computed by using a two-way, paired t-test (paired by trial) across all trials for each monkey separately. The global neural variability change was computed by averaging the variance across all recording areas for each trial (S2 Fig). Statistics for the regional neural variability change were corrected for multiple comparisons using an FDR-corrected threshold of $p < 0.05$.

## Spiking data: Task versus rest state correlation analysis

Neural correlations for spiking data using the same preprocessing steps mentioned above for spike rate variability analysis. Specifically, the mean-evoked response across all trials for each task condition was removed from each trial. Spike count correlations were then computed across trials, using groups of 25 trials as described above (Fig 2E and 2F).

The difference in task versus rest neural correlations was calculated using a two-way, paired t-test (paired by each bin of 25 trials) for each subject separately using Fisher's $z$-transformed correlation values. The global neural correlation change was computed by averaging the Fisher's $z$-transformed correlation values between all pairs of cortical regions, and comparing the averaged task versus rest correlation for each bin (Fig 2D). Statistics for the pairwise neural correlation change (Fig 2E–2G) were corrected for multiple comparisons using an FDR-corrected threshold of $p < 0.05$.

In addition, we computed the spike count correlation during the ITI and task cue period separately within trial (across time points) (S2 Fig). This analysis demonstrated that moment-to-moment correlations (rather than trial-to-trial correlations) were also quenched from rest to task periods, suggesting that correlation quenching also occurs at faster timescales. The statistical difference in task versus rest neural correlations was computed by using a two-way, paired t-test (paired by trial) across all trials for each monkey separately. The global neural

correlation change was computed by averaging the correlation across all pairs of recording areas for each trial (S2 Fig).

## fMRI: Data and paradigm

The present study was approved by the Rutgers University institutional review board. Data were collected as part of the Washington University-Minnesota Consortium of the Human Connectome Project (HCP) [34]. A subset of data ($n$ = 352) from the HCP 1200 release was used for empirical analyses. Specific details and procedures of subject recruitment can be found in [34]. The subset of 352 participants was selected based on: quality control assessments (i.e., any participants with any quality control flags were excluded, including 1) focal anatomical anomaly found in T1w and/or T2w scans, 2) focal segmentation or surface errors, as output from the HCP structural pipeline, 3) data collected during periods of known problems with the head coil, 4) data in which some of the FIX-ICA components were manually reclassified; low-motion participants (i.e., exclusion of participants that had any fMRI run in which more than 50% of TRs had greater than 0.25mm framewise displacement); removal according to family relations (unrelated participants were selected only, and those with no genotype testing were excluded). A full list of the 352 participants used in this study will be included as part of the code release.

All participants were recruited from Washington University in St. Louis and the surrounding area. We split the 352 subjects into two cohorts of 176 subjects: an exploratory cohort (99 females) and a replication cohort (84 females). The exploratory cohort had a mean age of 29 years of age (range = 22–36 years of age), and the replication cohort had a mean age of 28 years of age (range = 22–36 years of age). All subjects gave signed, informed consent in accordance with the protocol approved by the Washington University institutional review board. Whole-brain multiband echo-planar imaging acquisitions were collected on a 32-channel head coil on a modified 3T Siemens Skyra with TR = 720 ms, TE = 33.1 ms, flip angle = 52°, Bandwidth = 2,290 Hz/Px, in-plane FOV = 208x180 mm, 72 slices, 2.0 mm isotropic voxels, with a multiband acceleration factor of 8. Data for each subject were collected over the span of two days. On the first day, anatomical scans were collected (including T1-weighted and T2-weighted images acquired at 0.7 mm isotropic voxels) followed by two resting-state fMRI scans (each lasting 14.4 minutes), and ending with a task fMRI component. The second day consisted with first collecting a diffusion imaging scan, followed by a second set of two resting-state fMRI scans (each lasting 14.4 minutes), and again ending with a task fMRI session.

Each of the seven tasks was collected over two consecutive fMRI runs. The seven tasks consisted of an emotion cognition task, a gambling reward task, a language task, a motor task, a relational reasoning task, a social cognition task, and a working memory task. Briefly, the emotion cognition task required making valence judgements on negative (fearful and angry) and neutral faces. The gambling reward task consisted of a card guessing game, where subjects were asked to guess the number on the card to win or lose money. The language processing task consisted of interleaving a language condition, which involved answering questions related to a story presented aurally, and a math condition, which involved basic arithmetic questions presented aurally. The motor task involved asking subjects to either tap their left/right fingers, squeeze their left/right toes, or move their tongue. The reasoning task involved asking subjects to determine whether two sets of objects differed from each other in the same dimension (e.g., shape or texture). The social cognition task was a theory of mind task, where objects (squares, circles, triangles) interacted with each other in a video clip, and subjects were subsequently asked whether the objects interacted in a social manner. Lastly, the working memory task was a variant of the N-back task.

Further details on the resting-state fMRI portion can be found in [60], and additional details on the task fMRI components can be found in [30]. All fMRI results reported in the main article reflect results found with the first cohort of subjects. Independent replication of these effects are reported in S4 Fig with the replication cohort.

## fMRI: Preprocessing

Minimally preprocessed data for both resting-state and task fMRI were obtained from the publicly available HCP data. Minimally preprocessed surface data was then parcellated into 360 brain regions using the [61] atlas. We performed additional standard preprocessing steps on the parcellated data for resting-state fMRI and task state fMRI to conduct neural variability and FC analyses. This included removing the first five frames of each run, de-meaning and detrending the time series, and performing nuisance regression on the minimally preprocessed data [36]. Nuisance regression removed motion parameters and physiological noise. Specifically, six primary motion parameters were removed, along with their derivatives, and the quadratics of all regressors (24 motion regressors in total). Physiological noise was modeled using aCompCor on time series extracted from the white matter and ventricles [62]. For aCompCor, the first 5 principal components from the white matter and ventricles were extracted separately and included in the nuisance regression. In addition, we included the derivatives of each of those components, and the quadratics of all physiological noise regressors (40 physiological noise regressors in total). The nuisance regression model contained a total of 64 nuisance parameters. This was a variant of previously benchmarked nuisance regression models reported in [36].

We excluded global signal regression (GSR), given that GSR artificially induces negative correlations [57,63], which would bias analyses of the difference of the magnitude of correlations between rest and task. We included aCompCor as a preprocessing step here given that aCompCor does not include the circularity of GSR (regressing out some global gray matter signal of interest) while including some of the benefits of GSR (some extracted components are highly similar to the global signal) [64]. This logic is similar to a recently-developed temporal-ICA-based artifact removal procedure that seeks to remove global artifact without removing global neural signals, which contains behaviorally relevant information such as vigilance [65,66]. We extended aCompCor to include the derivatives and quadratics of each of the component time series to further reduce artifacts. Code to perform this regression is publicly available online using python code (version 2.7.15) (https://github.com/ito-takuya/fmriNuisanceRegression).

Task data for task FC analyses were additionally preprocessed using a standard general linear model (GLM) for fMRI analysis. For each task paradigm, we removed the mean evoked task-related activity for each task condition by fitting the task timing (block design) for each condition using a finite impulse response (FIR) model [27]. (There were 24 task conditions across seven cognitive tasks.) We used an FIR model instead of a canonical hemodynamic response function given recent evidence suggesting that the FIR model reduces both false positives and false negatives in the identification of FC estimates [27]. This is due to the FIR model's ability to flexibly fit the mean task-evoked response across all blocks. Removing the mean-evoked response of a task condition (i.e., main effect of task) is critical to isolate the spontaneous neural activity (and similarly the background connectivity [35]). Importantly, this procedure is standard when performing in spike count correlations [11,27]. Analogous statistical preprocessing steps were critical when comparing neural correlation measures across human fMRI data and NHP spiking data.

FIR modeled task blocks were modeled separately for task conditions within each of the seven tasks. Thus, the mean task-evoked activation was differentially accounted for according to each specific task condition. In particular, two conditions were fit for the emotion cognition task, where coefficients were fit to either the face condition or shape condition. For the gambling reward task, one condition was fit to trials with the punishment condition, and the other condition was fit to trials with the reward condition. For the language task, one condition was fit for the story condition, and the other condition was fit to the math condition. For the motor task, six conditions were fit: (1) cue; (2) right hand trials; (3) left hand trials; (4) right foot trials; (5) left foot trials; (6) tongue trials. For the relational reasoning task, one condition was fit to trials when the sets of objects were matched, and the other condition was fit to trials when the objects were not matched. For the social cognition task, one condition was fit if the objects were interacting socially (theory of mind), and the other condition was fit to trials where objects were moving randomly. Lastly, for the working memory task, 8 conditions were fit: (1) 2-back body trials; (2) 2-back face trials; (3) 2-back tool trials; (4) 2-back place trials; (5) 0-back body trials; (6) 0-back face trials; (7) 0-back tool trials; (8) 0-back place trials. Since all tasks were block designs, each time point for each block was modeled separately for each task condition (i.e., FIR model), with a lag extending up to 25 TRs after task block offset.

## fMRI: Task state activation analysis

We performed a task GLM analysis on fMRI task data to evaluate the task-evoked activity. The task timing for each of the 24 task conditions was convolved with the SPM canonical hemodynamic response function to obtain task-evoked activity estimates for each task condition separately [67]. FIR modeling was not used when modeling task-evoked activity. Coefficients were obtained for each parcel in the Glasser et al. (2016) cortical atlas for each of the 24 task conditions.

## fMRI: Task state versus resting-state variability analysis

To compare task-state versus resting-state variability, we regressed out the exact same task design matrix used on task-state regression on resting-state data. This was possible given that the number of timepoints of the combined resting-state scans in the HCP data set exceeded the number of timepoints of the combined task-state scans (4800 resting-state TRs > 3880 task-state TRs). This step was to ensure that any spurious change induced through the removal of the mean task-evoked response would also induce spurious changes in the resting-state data. However, results were qualitatively identical without the regression of the task design matrix on resting-state data.

After task regression, we obtained the residual time series for both resting-state and task state fMRI data. We then z-normalized each task run with zero-mean and unit variance such that we could appropriately compare the neural variability of task blocks across different runs. We emphasize that task activation regression (removal of the mean task-evoked response) was removed *prior* to z-scoring the time series. (Additionally, S5 Fig shows results without *z*-normalization, and the results are qualitatively identical.) This enabled us to evaluate whether the variability during task blocks significantly decreased relative to inter-block intervals by evaluating the variance of task blocks relative to 1. We then extracted the time series variance during task blocks, and then averaged the variance across all task conditions to obtain our statistic of task-evoked neural variability. To identify the resting-state neural variability, we applied the same exact procedure to resting-state time series using the task-state design matrix. A sanity check for our analysis was that the 'intrinsic-state' neural variability is close to 1 (given that the time series was normalized to have unit variance), while the task-state neural variability is

significantly less than 1 (Fig 3A). This ensured that variability measures were not biased by the normalization step.

We compared the neural variability of the entire brain during task state periods versus resting-state periods. For each subject, we computed the variance during task and resting state separately, and then averaged across all brain regions. This resulted in two values per subject, representing task state and resting-state variability. We then performed a two-way group paired t-test across subjects to assess statistical significance (Fig 3A). We also computed the task state versus resting-state difference in neural variability for each brain region separately (Fig 3B). We corrected for multiple comparisons using an FDR-corrected threshold of $p < 0.05$ (Fig 3B and 3C). Cortical surface visualizations were constructed using Connectome Workbench (version 1.2.3) [34].

### fMRI: Task state versus resting-state correlation analysis

We compared task-state versus resting-state FC (i.e., neural correlations), after performing the exact same preprocessing steps as mentioned above. Results without z-normalization (and using covariance rather than correlations) on the task and rest residual time series are reported in S5 Fig.

We computed the correlation between all pairs of brain regions for each task condition during task block periods. We then averaged the Fisher's $z$-transformed correlation values across all task conditions to obtain a general task state FC matrix (Fig 3E). We repeated the same procedure (i.e., using the same task-timed blocks) on resting-state FC to obtain an equivalent resting-state FC matrix for each subject (Fig 3D). We directly compared task-state FC to resting-state FC by performing two-way group paired t-tests for every pair of brain regions using the Fisher's $z$-transformed correlation values. Statistical significance was assessed using an FDR-corrected threshold of $p < 0.05$ (Fig 3F). To compare the average global correlation during task state and resting state, we computed the average correlation between all pairs of brain regions during task and resting-state, performing a group paired t-test (Fig 3G). To compare the average global connectivity profile of every brain region [68], we computed the average Fisher $z$-transformed correlation of a single region to all other brain regions during task and rest and performed a two-way group paired t-test between task and rest (Fig 3H and 3I). Statistical significance was assessed using an FDR-corrected threshold of $p < 0.05$.

### fMRI: Task state versus resting-state variability/correlation analysis without task regression

To compare task-state versus resting-state variability/correlations *without* regressing out task effects using FIR [27], we calculated the variance/correlations for each time point across blocks. This approach is similar to previous studies that measured variability changes after task/stimulus onset [2,4]. Importantly, because variance/correlations explicitly account for the mean across a sample, and variance/correlations are computed for each time point separately, this approach accurately accounts for task-locked effects.

Statistics (i.e., variance/correlations) were calculated across blocks at each time point, for each condition separately. To accurately compare task-state to resting-state statistics, we computed cross-block statistics for rest data using the same task block design (i.e., sham/control blocks). This controlled for the number of task blocks and temporal spacing between blocks. We included the first 15 time points following block onset for both the rest and task data. Thus, any task blocks that contained fewer than 15 time points were excluded. This was performed for all ROIs for every subject. Summary statistics were aggregated across ROIs, task conditions and time points (within rest or task states) and visualized in Fig 4.

(For this analysis, we used minimally preprocessed data (from the HCP). Additional nuisance regression was performed for both rest and task data as described above, excluding task activation regression.)

## Information-theoretic analysis

We evaluated the information-theoretic relevance of rest and task states by characterizing the dimensionality of neural activity. To estimate the statistical dimensionality of neural data, we used the 'participation ratio', as previously described in [39]. We first obtain the covariance matrix $W$ of activity for rest and task states separately. We then calculated

$$dim_W = \frac{(\sum_{i=1}^{m} \lambda_i)^2}{\sum_{i=1}^{m} \lambda_i^2} \tag{2}$$

Where $dim_W$ corresponds to the statistical dimensionality of $W$, and $\lambda_i$ corresponds to the eigenvalues of the covariance matrix $W$. Intuitively, this is related to finding the number of components needed to explain variance greater than some fixed threshold, with more needed components reflecting a higher dimensionality of the data.

For human fMRI data, we estimated the task-state and resting-state dimensionality by calculating the whole-brain covariance matrix for each state. For task state this was done by estimating the covariance matrix using task block periods. For resting state this was done by calculating the covariance matrix across the equivalently lengthed resting-state periods (using the same data in the FC analysis above). We applied Eq 2 to the task-state and resting-state covariance matrix for each subject. Finally, we applied a two-way, group paired t-test comparing the dimensionality of task-state activity to resting-state activity (Fig 5A). We replicated this finding in the replication cohort. In addition, we performed this analysis for each fMRI task separately (S10 Fig).

For NHP spiking data, we estimated the task (task cue period) and rest (ITI) dimensionality by calculating the covariance matrix between all pairs of population recordings. We then applied Eq 2 to task and rest periods for each covariance matrix. (Each covariance matrix was calculated using bins of 25 consecutive trials.) Finally, we applied a two-way group paired t-test (across bins) comparing the dimensionality of task activity to rest activity (Fig 5B). We replicated this effect in the held-out second monkey.

## Spiking model: Estimating the transfer function of a neural population with a balanced spiking model

Our goal was to evaluate the effects of evoked activity across large neural populations, rather than within populations. Thus, we first estimated the transfer function of a neural population using a previously established balanced neural spiking model, with 4000 excitatory and 1000 inhibitory units [7]. All parameters are taken directly from [7] with the description paraphrased below. Units within the network were modeled as leaky integrate-and-fire neurons whose membrane voltages obeyed the equation

$$\frac{dV}{dt} = \frac{1}{\tau}(\mu - V) + I_{syn} \tag{3}$$

where $\tau$ indicates the membrane time constant, $\mu$ is the bias term, and $I_{syn}$ is the synaptic input. When neurons reached $V_{th} = 1$, a spike was emitted, and voltages were reset to $V_{re} = 0$ for an absolute refractory period of 5ms. $\tau$ was 15ms and 10ms for excitatory and inhibitory neurons, respectively. For excitatory neurons, $\mu$ was randomly sampled from a uniform

distribution between 1.1 and 1.2. For inhibitory neurons, $\mu$ was randomly sampled from a uniform distribution between 1 and 1.05.

Synapses to a neuron were modeled as the sum of excitatory and inhibitory synaptic trains $x_E$ and $x_I$, respectively, and was calculated as the normalized difference of exponentials describing the synaptic rise and decay times caused by each presynaptic event. This effectively captured the weighted effect of all presynaptic neurons to a target neuron, and specifically obeyed the equations

$$I_{y,syn} = x_E(t) + x_I(t) \tag{4}$$

$$x_Z(t) = \frac{x_{Z,decay} - x_{Z,rise}}{\tau_{Z,decay} - \tau_{Z,rise}}, \ Z \in \{E, I\} \tag{5}$$

where the synaptic rise and decay of $x_E$ and $x_I$ was modeled as the first order differential equations

$$\frac{dx_{Z,decay}}{dt} = \sum_j J_{ij} s_j - \frac{x_{Z,decay}}{\tau_{Z,decay}} \tag{6}$$

$$\frac{dx_{Z,rise}}{dt} = \sum_j J_{ij} s_j - \frac{x_{Z,rise}}{\tau_{Z,rise}} \tag{7}$$

$J_{ij}$ refers to the synaptic weight from neuron j to i, $s_j$ indicates whether neuron j emitted a spike. Synaptic rise times were the same for excitatory and inhibitory neurons, with $\tau_{E,rise} = \tau_{I,rise} = 1ms$, while $\tau_{E,decay} = 3ms$ and $\tau_{I,decay} = 2ms$. Connection probabilities $p^{xy}$ from neurons in population $y$ to $x$ were $p^{EI} = p^{IE} = p^{II} = 0.5$, and on average, $p^{EE} = 0.2$. However, if two neurons were both excitatory and belonged to the same cluster, the connection strength was multiplied by 1.9. (We employed only the homogenous clustered networks, as described by [7]; $J^{EE} = 0.024$, $J^{EI} = -0.045$, $J^{IE} = 0.014$, and $J^{II} = -0.057$.) Excitatory stimulation was performed by increasing $\mu$ to the first 400 excitatory neurons from 0.05 to 0.5 in 0.05 increments. Inhibitory stimulation was performed by decreasing $\mu$ by 0.05 to 0.5 in 0.05 increments to 400 inhibitory neurons.

To estimate the population transfer function, we simulated 30 trials lasting 2s each at each stimulation amplitude. Spike train statistics were estimated across trials in 50ms sliding windows with 10ms shifts. Only excitatory neurons were included when calculating the population spike train statistics (i.e., mean and variance at each stimulation amplitude).

Model code was originally adapted from [7], and was simulated with Julia (version 1.1.1).

## Model: One-dimensional minimal network model

We use the simplest model to mathematically characterize the relationship between evoked activity and neural variability: a one-dimensional mean-field model. We used Wilson-Cowan-type firing rate dynamics to simulate neural population activity [69]. Specifically, our population's activity obeyed the equation

$$\tau_i \frac{dx_i}{dt} = -x_i + f(w_{ii} x_i + b_i + s_i + I) \tag{8}$$

where $x_i$ denotes the firing rate (or a measure of activity), $\tau_i = 0.1$ denotes the time constant, $w_{ii} = 1$ refers to local coupling (auto-correlation), $b_i = -0.5$ refers to the input threshold for optimal activity (or a bias term), $s_i$ refers to the evoked stimulation ($s_i = 0$ for intrinsic activity),

*I* refers to background spontaneous activity sampled from a Gaussian distribution with mean 0 and standard deviation 0.25, and *f* is a sigmoid input-output activation function, which is defined as

$$f(x) = \frac{1}{1 + e^{-k*x}}$$

(9)

where *k* = 1. Numerical simulations were computed using a Runge-Kutta second order method with a time step of dt = 10ms [70]. We simulated neural population activity injecting a fixed input (boxcar input) with amplitudes ranging from $s_i \in [-5,5]$ in 0.01 increments (Fig 7C). Neural variability for each input strength was calculated using the standard deviation of the time series following the input onset and preceding input offset. Each trial was run for 20 seconds. Fig 7C was generated using input amplitudes of $s_i \in \{-3,0,3\}$.

To visualize the full dynamics of our single neural population, we visualized the one-dimensional phase space (i.e., flow field on a line) [43]. In particular, we calculated the flow field by plotting $\dot{x}$ (i.e., $\frac{dx}{dt}$) as a function of $x_i$ (Eq 8). Notably, fixed point attractors (equilibrium states) are defined where $\dot{x} = 0$ (Fig 6B).

## Model: Two-dimensional minimal network model

To characterize the effects of evoked activity on neural correlations, we use a two-dimensional neural population model. We extended the one-dimensional network model to include two neural populations. The network dynamics obeyed the equations

$$\tau_1 \frac{dx_1}{dt} = -x_1 + f(w_{11}x_1 + w_{21}x_2 + b_1 + s_1 + I_1)$$

(10)

$$\tau_2 \frac{dx_2}{dt} = -x_2 + f(w_{22}x_2 + w_{12}x_1 + b_2 + s_2 + I_2)$$

(11)

where $x_1$ and $x_2$ describe the activity of each population, and all other variables are as described above. Inter-regional coupling was set to be greater than local coupling, given evidence from previous studies that global coupling is greater than local coupling [48,71,72]. Specific network parameters for this network model were: $w_{11} = w_{22} = 2$, $w_{12} = w_{21} = 4$, $b_1 = b_2 = -3$, $\tau_1 = \tau_2 = 0.1$. $I_1$ and $I_2$ were sampled from a Gaussian distribution with mean 0 and standard deviation 1. For this network model, we decreased the slope of the sigmoid $k = 0.5$ to allow for a larger dynamic, linear response range. The full model code is publicly available: https://github.com/ito-takuya/corrquench.

To quantify the relationship between evoked activity and neural correlations, we systematically simulated the network under different stimulation states (input strengths). Using the same methods as above, we simulated network activity for 50 seconds. We injected fixed input into both neural populations with amplitudes ranging from $s_i \in [-5,5]$ in 0.01 increments (Fig 7E). Notably, given that the injected stimulation is uncorrelated (due to 0-variance in a fixed input), it is non-trivial that the FC between two nodes would change in response to different inputs. Neural correlations were calculated using a Pearson correlation of the two time series following input onset and preceding input offset.

The use of a minimal model constrained our network to two dimensions. This allowed us to leverage dynamical systems tools to visualize the flow field in the two-dimensional phase plane. To identify the fixed point attractors, we first calculated the nullclines for $x_1$ and $x_2$. Nullclines are defined as the values of $x_1$ and $x_2$ such that $\dot{x_2} = 0$ and $\dot{x_1} = 0$, respectively. The fixed points lie at the intersection of the nullclines. For our particular system of equations, the

nullclines of $x_1$ and $x_2$ were defined, respectively, as

$$x_2 = \frac{f^{-1}(-x_1) - w_{11}x_1 - b_1 - s_1}{w_{21}} \tag{12}$$

$$x_1 = \frac{f^{-1}(-x_2) - w_{22}x_2 - b_2 - s_2}{w_{12}} \tag{13}$$

where parameters are identical to those used in Eqs 10 and 11. However, the background noise, parameter $I$, was removed when calculating the nullclines. Fixed point attractors (equilibrium states) are defined where $\dot{x} = 0$ (Fig 8B). The full flow field was obtained by applying the system of equations (Eqs 10 and 11) to every point in the phase space (e.g., all values of $x_1$ and $x_2$).

## Model: Evaluating fixed point attractor dynamics and the characteristic time scale

Our models accurately demonstrated that evoked activity decreased the neural variability and correlations from a stochastic dynamical network model. Since our network model was governed by firing rate equations which provided us full access to the system's dynamics, we sought to link dynamical mechanisms (in the absence of spontaneous activity) with changes in the descriptive statistics. Such an analysis would provide us with a mechanistic understanding between descriptive neural statistics used in empirical data analysis and the governing neural dynamics.

To understand how attractor dynamics influenced simulated activity in a network model, we characterized the dynamics around the network's fixed point attractor. Specifically, we performed a linear stability analysis around the fixed point (i.e., the equilibrium level of activity the system is drawn to during a particular state or input, e.g., Fig 7B) in both the one-dimensional and two-dimensional network models. In the one-dimensional case, this analysis is equivalent to evaluating the first derivative of Eq 8 at the fixed point (e.g., the slope of the line at the starred locations in Fig 7B). We then calculate the *characteristic time scale* T at the fixed point $x^*$ (in one-dimension) with the equation

$$T = \frac{1}{|f\prime(x^*)|} \tag{14}$$

where $f$ represents Eq 8 [43]. The characteristic time scale captures the speed with which the system approaches the fixed point attractor. We calculated the characteristic time scale across the same range of evoked stimulation strengths as in the neural variability analysis. Fixed points were computed numerically by running the network model until it reached a steady state in the absence of noise/spontaneous activity.

The characteristic time scale is an established measure for one-dimensional systems. However, we sought to extend the characteristic time scale beyond a single dimension to evaluate shifting attractor dynamics in higher dimensions. We first performed a linear stability analysis in two dimensions by evaluating the Jacobian matrix for our two-dimensional system at the fixed point $(x^*_1, x^*_2)$

$$J(x_1^*, x_2^*) = \begin{pmatrix} \dfrac{df_1}{dx_1} & \dfrac{df_1}{dx_2} \\ \dfrac{df_2}{dx_1} & \dfrac{df_2}{dx_2} \end{pmatrix} \tag{15}$$

Where $f_1$ and $f_2$ refer to the equations governing neural populations 1 and 2 (Eqs 10 and 11, respectively). For our particular system of equations, the Jacobian was calculated as

$$J(x_1^*, x_2^*) = \begin{pmatrix} (-1 + f'(w_{11}x_1 + w_{21}x_2 + b_1 + s_1))\dfrac{1}{\tau_1} & (f'(w_{11}x_1 + w_{21}x_2 + b_1 + s_1))\dfrac{1}{\tau_1} \\ (f'(w_{22}x_2 + w_{12}x_1 + b_2 + s_2))\dfrac{1}{\tau_2} & (-1 + f'(w_{22}x_2 + w_{12}x_1 + b_2 + s_2))\dfrac{1}{\tau_2} \end{pmatrix} \quad (16)$$

For each input strength (i.e., differing evoked states), we evaluated the Jacobian at the fixed point attractor. We then calculated the two eigenvalues (denoted $\lambda_1$ and $\lambda_2$) and eigenvectors (denoted $\mathbf{v}_1$ and $\mathbf{v}_2$) of the Jacobian using an eigendecomposition. To calculate the generalized characteristic time scale in two dimensions, we first calculated the linear combination of the eigenvectors weighted by the real eigenvalues, and computed the magnitude of the vector, such that

$$v_{sum}(x, y) = re(\lambda_1)v_1 + re(\lambda_2)v_2 \quad (17)$$

We then define the two dimensional characteristic time scale $T$ as the reciprocal of the magnitude of $v_{sum}(x,y)$, such that

$$T = \frac{1}{|\sqrt{x^2 + y^2}|} \quad (18)$$

We calculated $T$ for a range of values $s_1, s_2 \in [-5,5]$ in 0.01 increments, and correlated $T$ across all values of $s_1$ and $s_2$ with the corresponding neural correlations [43].

## Model: 300 unit firing rate model

To verify the findings observed in our minimal models would scale to larger networks (one and two-dimensional models), we included a 300 region mean-field firing rate model. We chose 300 regions given that most whole-brain human atlases contain 200–400 cortically defined parcels [13,61,73]. Our model followed the same equations as in our minimal model, though inter-area weights were appropriately scaled relative regional self-coupling parameters. Specifically, the network dynamics obeyed the equations

$$\tau_i \frac{dx_i}{dt} = -x_1 + f\left(w_{ii}x_i + \sum_{j \neq i}^{300} w_{ji}x_j + b_i + s_i + I_i\right) \quad (19)$$

where $x_i$ describes the activity of each population, and all other variables are as described above. Inter-regional coupling was set to be greater than local coupling (2:1 ratio), given evidence from previous studies that global coupling is greater than local coupling [48,71,72]. Specific parameters for this network model were specified such that: $w_{ii} = 1$, the mean of inter-region coupling parameters was $\sum_{j \neq i}^{300} w_{ji} = 2$, $b_i = -2$, $\tau_i = 0.1$. $I_i$ was sampled from a Gaussian distribution with mean 0 and standard deviation 1. $s_i = 0$ during rest state and $s_i = 1$ during task state.

In total, we ran simulations for two classes of network models: a network with random connections and a network with clustered communities (S11 Fig). For the random network, we randomly sampled connections with 20% probability rate between all pairs of regions. For the clustered network model, we generated 10 communities of 30 nodes each. Regions within the community had a 20% probability rate for establishing a connection. Between-community connections had a 3% probability rate for establishing a connection.

For each class of network (random or clustered), we weighted connections with either positive weights only (i.e., only E connections) or both positive and negative weights (i.e., both E and I connections). For the E-only network, weights were sampled from a normal distribution with parameters $\mu = 1$, $\sigma = 0.2$. For the network with both E and I weights (80% E, 20% I), weights were sampled from a normal distribution with parameters $\mu = 1$, $\sigma = 1.2$.

Both the rest and task state simulation was run for 10 seconds each and sampled at 100ms. For each group analysis (S11 Fig), we simulated 20 subjects worth of data. The full model code is publicly available, and was written in python 3.7.3.

## Model: Simulating fMRI BOLD activity

We used the above model to simulate fMRI BOLD activity to demonstrate that changes in neural variability and correlations would extend to fMRI BOLD dynamics (S6 and S7 Figs). Neural activity generated from our model simulations was transformed to fMRI BOLD activity using the Balloon-Windkessel model, a nonlinear transformation from neural activity to the fMRI BOLD signal [44,74]. Notably, the transformation assumes a nonlinear transformation of the normalized deoxyhemoglobin content, normalized blood inflow, resting oxygen extraction fraction, and the normalized blood volume. All state equations and biophysical parameters were taken directly from [44] (Eqs 4 and 5). The Balloon-Windkessel model was implemented in Python (version 2.7.13), and the implementation code has been made publicly available on GitHub (https://github.com/ito-takuya/HemodynamicResponseModeling).

## Ethics statement

All procedures performed on the two rhesus monkeys (one male, one female) followed the guidelines of the Massachusetts Institute of Technology Committee on Animal Care and the National Institutes of Health.

All human participants included in this study gave signed, informed consent in accordance with the protocol approved by the Washington University institutional review board.

## Supporting information

**S1 Fig. Replication analysis for the excluded NHP subject.** This figure is organized identically to Fig 2, but using data from a replication subject. We find nearly identical patterns between the exploratory and replication subjects, with the exception that we did not replicate any correlation increases. **a)** Mean-field spike recordings from six different cortical regions fed into our analyses. **b)** As in our empirical fMRI data set, we calculated the global variability across task and rest states (estimated using the standard deviation across trials). **c)** We then calculated the global neural correlation (i.e., the spike count correlation across trials) for task and rest states between all pairs of recorded brain regions. (Spike rates were averaged within each cortical area.) **d-f)** For each pair of brain regions, we visualized the correlation matrices between each recording site for the averaged rest, task, and the differences between task versus rest state spike count correlations. For panels **d-f**, plots were thresholded and tested for multiple comparisons using an FDR-corrected $p < 0.05$ threshold. Boxplots indicate the interquartile range of the distribution, dotted black line indicates the mean, grey line indicates the median, and the distribution is visualized using a swarm plot.
(EPS)

**S2 Fig. Neural correlations and variability are quenched within trials from rest to task intervals.** We analyzed the variability across time points (within trial) during ITIs and task cue periods to evaluate whether correlation and variability quenching also occurred on a moment-

to-moment basis (i.e., faster timescale). Task cue intervals and ITIs were matched to have equivalent time points on a trial-by-trial basis. **a1,a2)** Global variability across the two states (estimated using the variance across time points) between task and rest state windows. **b1,b2)** We then calculated the global spike count correlation between the exact same task cue intervals with equivalent rest intervals between all pairs of recorded brain regions. (Spike rates were averaged within each cortical area.) **c1,c2)** We also calculated the global firing rate (averaged across all recording areas) during the task interval and rest interval. **d1-f1,d2-f2)** For each pair of brain regions, we visualize the spike count correlation matrices between each recording site for the averaged rest, task, and the differences between task versus rest state spike count correlation. For panels **d-f**, plots were thresholded and tested for multiple comparisons using an FDR-corrected p<0.05 threshold. Boxplots indicate the interquartile range of the distribution, dotted black line indicates the mean, grey line indicates the median, and the distribution is visualized using a strip plot.
(TIF)

**S3 Fig. Task-evoked activity is negatively correlated with variability and correlations across regions in fMRI data. a)** We replicated a previous result [2], demonstrating that regions that activated more during tasks tend to decrease their BOLD variability more during task states. **b)** We extended those results to evaluate the relationship between task-evoked activity and FC across regions. We found that regions that activated more during tasks tend to decrease their global functional FC accordingly during task states. Scatter plots reflect each parcel in the Glasser atlas [61], and are colored according to network affiliation [14]. Best fit lines were estimated using linear regression, but correlations were calculated using a non-parametric rank correlation. **c,d)** Replication of panels a,b, respectively using the replication cohort of subjects. Statistics were calculated using the same steps as in [2]. To calculate the averaged regional task activation, we first performed a group t-test for each task against 0, took the absolute value of the t-statistic, and then averaged across tasks. To calculate the averaged regional FC and SD, we performed a group t-test against 0 for each region. We then correlated these values across regions to measure the relationship between activity and FC, and activity and SD.
(TIF)

**S4 Fig. Replication data set: Variability and correlations decrease during task states in human fMRI data.** We successfully replicated results from Fig 3 using our held-out cohort of 176 subjects. **a)** We first compared the global variability during task and rest states, which is averaged across all brain regions, and then **b)** computed the task- versus rest-state variability for each brain region. **c)** Scatter plot depicting the variance of each parcel during task states (y-axis) and rest states (x-axis). Dotted grey line denotes no change between rest and task states. **d)** We next compared the correlation matrices for resting state blocks with **(e)** task state blocks, and **(f)** computed the task- versus rest-state correlation matrix difference. **g)** We found that the average FC between all pairs of brain regions is significantly reduced during task state. **h)** We found that the average correlation for each brain region, decreased for each brain region during task state. **i)** Scatter plot depicting the FC (correlation values) of each pair of parcels during task states (y-axis) and rest states (x-axis). Dotted grey line denotes no change between rest and task states. For panels **b-f,** and **h,** plots were tested for multiple comparisons using an FDR-corrected $p<0.05$ threshold. Boxplots indicate the interquartile range of the distribution, dotted black line indicates the mean, grey line indicates the median, and the distribution is visualized using a swarm plot.
(TIF)

**S5 Fig. Non-normalized data using variance and covariance, using the full set of 352 subjects. Variance and covariance decreased during task states in human fMRI data.** We successfully replicated results from Fig 3 using, but without z-normalizing the time series (and using covariance instead of correlation). The combination of reduced correlations (Fig 3) and covariance measures suggested that shared signal dynamics is reduced from task to rest [31–33]. **a)** We first compared the global variability during task and rest states, which is averaged across all brain regions, and then **b)** computed the task- versus rest-state variability for each brain region. **c)** Scatter plot depicting the variance of each parcel during task states (y-axis) and rest states (x-axis). Dotted grey line denotes no change between rest and task states. **d)** We next compared the covariance matrices for resting state blocks with **(e)** task state blocks, and **(f)** computed the task- versus rest-state covariance matrix difference. **g)** We found that the average covariance between all pairs of brain regions is significantly reduced during task state. **h)** We found that the average covariance for each brain region, decreased for each brain region during task state. **i)** Scatter plot depicting the FC (covariance values) of each pair of parcels during task states (y-axis) and rest states (x-axis). Dotted grey line denotes no change between rest and task states. For panels **b-f,** and **h**, plots were tested for multiple comparisons using an FDR-corrected $p < 0.05$ threshold. Boxplots indicate the interquartile range of the distribution, dotted black line indicates the mean, grey line indicates the median, and the distribution is visualized using a swarm plot.
(TIF)

**S6 Fig. Task-state neural variability reduction is preserved in the BOLD signal in the neural mass model. a)** We simulated the neural mass model under the same three stimulus conditions (de-activated, baseline, and activated states) as in Fig 7A. **b)** We subsequently applied the Balloon-Windkessel transformation to the simulated neural activity, a nonlinear transformation from neural activity to the fMRI BOLD signal [44]. Notably, the transformation assumes a nonlinear transformation of the normalized deoxyhemoglobin content, normalized blood inflow, resting oxygen extraction fraction, and the normalized blood volume. All BOLD signals were de-meaned such that it is possible to visually compare the time series variance of each stimulus condition. **c)** We simulated BOLD activity under a range of stimulus conditions and calculated the standard deviation of each time series. **d)** We calculated the rank correlation of the standard deviation of the BOLD signal across stimulus conditions with the characteristic time scale at each condition.
(TIF)

**S7 Fig. Task-state neural correlation reduction is preserved in the BOLD signal in the two-unit neural mass model. a-b)** Using the simulated the neural mass data in Fig 8, we applied the Balloon-Windkessel transform to convert our neural data into BOLD data [44]. **c)** We simulated BOLD activity under a range of stimulus conditions and calculated the neural correlation between the two units of each. **d)** We calculated the rank correlation of the neural correlation of the BOLD signal across stimulus conditions with the characteristic time scale at each condition.
(EPS)

**S8 Fig. fMRI variability reduction analysis for each of the 7 HCP tasks separately. a)** This panel is identical to the analysis performed in Fig 3A, except that it was performed on each HCP task separately. Global variability, averaged across all regions, was reduced for 4/7 of the HCP tasks. Global variability was not reduced for the Emotion and Motor tasks, though task-evoked activity was correlated with task-evoked variability reduction across space (see next panel). **b)** This panel is identical to the analysis performed in S3 Fig, except that the spatial

correlation was performed on each HCP task separately (and is visualized as a bar plot rather than a scatter plot). Regional task-evoked variability was significantly negatively correlated with the magnitude of task-evoked activation (absolute value) for 6/7 of the HCP tasks. All analyses (in panels a and b) were corrected for multiple comparisons using FDR correction. (*** = FDR-corrected $p<0.0001$; ** = FDR-corrected $p<0.01$; * = FDR-corrected $p<0.05$). Boxplots indicate the interquartile range of the distribution, dotted black line indicates the mean, and the distribution is visualized using a swarm plot.
(EPS)

**S9 Fig. Task versus rest fcMRI analysis for each of the 7 HCP tasks separately. a)** This panel is identical to the analysis performed in Fig 3G, except that it was performed on each HCP task separately. Whole-brain FC, averaged across all pairs of regions, was reduced for 7/7 of the HCP tasks. **b)** This panel is identical to the analysis performed in S3 Fig, except that the spatial correlation was performed on each HCP task separately (and is visualized as a bar plot). Regional task-evoked FC was significantly negatively correlated with the magnitude of task-evoked activation (absolute value) for 4/7 of the HCP tasks. All analyses (in panels A and B) were corrected for multiple comparisons using FDR correction. (*** = FDR-corrected $p<0.0001$; ** = FDR-corrected $p<0.01$; * = FDR-corrected $p<0.05$). **c)** Task- versus rest-state FC analysis for each of the 7 HCP tasks separately. (This figure is identical to Fig 3f, except that the statistics were performed on each task separately.) Though whole-brain FC differences from task to rest are different for each task, there are mostly FC decreases during task state relative to rest state. Boxplots indicate the interquartile range of the distribution, dotted black line indicates the mean, and the distribution is visualized using a swarm plot.
(TIF)

**S10 Fig. Task versus rest dimensionality comparison for each of the 7 HCP tasks separately. a)** This panel is identical to the analysis performed in Fig 5A, except that it was performed on each HCP task separately. Whole-brain dimensionality increased from rest to task states for each of the 7 HCP tasks. Boxplots indicate the interquartile range of the distribution, dotted black line indicates the mean, and the distribution is visualized using a swarm plot. (*** = FDR-corrected $p<0.0001$; ** = FDR-corrected $p<0.01$; * = FDR-corrected $p<0.05$)
(EPS)

**S11 Fig. Variability and correlations are quenched in large-scale network models (300 regions) with both random and clustered structural connections.** For each structural connectivity matrix, we randomly sampled synaptic weights from a normal distribution with either 100% E connections (given evidence that most long-range connections are excitatory, $\mu = 1.0$, $\sigma = 0.2$ [42], or 80% E and 20% I connections ($\mu = 1.0$, $\sigma = 1.2$). For each network model (4 in total), we simulated 20 subjects for 10 seconds each (100ms sampling rate). For simplicity, during the task state, all units were stimulated with a fixed input. **a)** Random structural connectivity matrix (20% connectivity density) for an example subject. **b)** The average across all pairwise correlations during the rest and task states for the network model with 80% E and 20% I connections. The rest state exhibits higher correlations than the task state. **c)** The variability (variance across time) averaged across brain regions during the rest and task states for the network model with 80% E and 20% I connections. The rest state exhibits higher variability than the task state. **d)** The task minus rest FC matrix (correlation difference) between all 300 regions. Correlations decreased from rest to task states. **e-g)** The same analyses as **b-d**, but using only excitatory connections only. **h)** Clustered structural connectivity matrix (10 communities, 20% within-community density, 3% out-of-community density). **i-k)** The same analyses as **b-d**, but using the clustered connectivity matrix with 80% E and 20% I connections. **l-**

**n)** The same analyses as **b-d**, but using the clustered connectivity matrix with 100% E connections. Boxplots indicate the interquartile range of the distribution, dotted black line indicates the mean, and the distribution is visualized using a swarm plot. Plots **d, g, k, n,** were corrected for multiple comparisons and thresholded using an FDR-corrected $p < 0.05$.
(TIF)

**S12 Fig. Rest (ITI) to task state (task cue) changes in fano factor analyzed for each neuron individually across the six cortical areas for the exploratory subject.** (This is *not* a mean-field analysis.) **a)** The distribution of fano factor across all neurons in FEF (from all recording sessions) for the rest (ITI) and task state (cue) periods. **b)** For each individual neuron in FEF, we calculated the change in fano factor from the rest to task state period. **c,d)** Same as **a, b**, but for PFC. **e,f)** Same as **a, b**, but for LIP. **g,h)** Same as **a, b**, but for IT. **i,j)** Same as **a, b**, but for MT. **k,l)** Same as **a, b**, but for V4.
(TIF)

**S13 Fig. Rest (ITI) to task state (task cue) changes in fano factor analyzed for each neuron individually across the six cortical areas for the replication subject.** (This is *not* a mean-field analysis.) **a)** The distribution of fano factor across all neurons in FEF (from all recording sessions) for the rest (ITI) and task state (cue) periods. **b)** For each individual neuron in FEF, we calculated the change in fano factor from the rest to task state period. **c,d)** Same as **a, b**, but for PFC. **e,f)** Same as **a, b**, but for LIP. **g,h)** Same as **a, b**, but for IT. **i,j)** Same as **a, b**, but for MT. **k,l)** Same as **a, b**, but for V4.
(TIF)

**S14 Fig. The average fano factor change from rest (ITI) to task (cue) periods for both the exploratory and replication NHP subjects. a)** Exploratory subject. **b)** Replication subject.
(EPS)

**S15 Fig. Scatter plot representations of averaged neural statistics (firing rate, variability, correlations, dimensionality) during rest and task state periods. All data are identical to those reported in Fig 2B–2D, Fig 5B, and S1B–S1D Fig, but are visualized as a scatter plot.** In each scatter plot, every point reflects the statistic (mean, variance, correlations, dimensionality) estimated across a bin of 25 contiguous trials. (Rest periods were defined as the ITI preceding the task cue onset.) Statistics were averaged across all recording sites, and included all recording sessions. **a)** The firing rate (averaged across six cortical areas) for task (y-axis) and rest (x-axis) states. **b)** The variance (averaged across six cortical areas) for task (y-axis) and rest (x-axis) states. **c)** Correlations (averaged across all pairwise correlations) for task (y-axis) and rest (x-axis) states. **d)** Dimensionality (i.e., participation ratio) of all six cortical areas during task (y-axis) and rest (x-axis) states. **e-h)** The same as **a-d**, but for the replication subject.
(TIF)

## Acknowledgments

We thank Drew B. Headley, Bart Krekelberg, Olaf Sporns, and Mauricio Delgado for helpful feedback on earlier versions of the manuscript.

## Author Contributions

**Conceptualization:** Takuya Ito, Michael W. Cole.

**Data curation:** Scott L. Brincat, Markus Siegel, Earl K. Miller.

**Formal analysis:** Takuya Ito.

**Funding acquisition:** Markus Siegel, Earl K. Miller, Michael W. Cole.

**Investigation:** Takuya Ito, Scott L. Brincat, Markus Siegel, Ravi D. Mill, Biyu J. He, Earl K. Miller, Horacio G. Rotstein, Michael W. Cole.

**Methodology:** Takuya Ito, Horacio G. Rotstein, Michael W. Cole.

**Resources:** Takuya Ito, Scott L. Brincat, Markus Siegel, Earl K. Miller.

**Software:** Takuya Ito.

**Supervision:** Horacio G. Rotstein, Michael W. Cole.

**Writing – original draft:** Takuya Ito, Michael W. Cole.

**Writing – review & editing:** Takuya Ito, Scott L. Brincat, Markus Siegel, Ravi D. Mill, Biyu J. He, Earl K. Miller, Horacio G. Rotstein, Michael W. Cole.

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
