## [Decision Letter · Decision Letter 0]

24 Feb 2020

Dear Mr. Ito,

Thank you very much for submitting your manuscript "Task-evoked activity quenches neural correlations and variability across cortical areas" for consideration at PLOS Computational Biology.

As with all papers reviewed by the journal, your manuscript was reviewed by members of the editorial board and by independent reviewers.

While the motivation and the effort put in the paper appear laudable, there appear to be serious issues with the choice of the model and the interpretation of the results. It's not clear to which extent it will be possible to address these issues without the paper becoming a completely different one, but still we want to give you the possibility to reconsider a massively revised version of the paper.

We cannot make any decision about publication until we have seen the revised manuscript and your response to the reviewers' comments. Your revised manuscript is also likely to be sent to reviewers for further evaluation.

Sincerely,

Daniele Marinazzo

Deputy Editor

PLOS Computational Biology

Reviewer's Responses to Questions

**Comments to the Authors:**

Reviewer #1: I think this is a nice paper and I recommend publication.

Reviewer #2: The authors analyse two different datasets (human fMRI and NHP electrophysiological data) for spontaneous and evoked activity under different tasks. They monitor the variability and correlations in the signal in rest and task conditions concluding that the task reduces the variability and the correlations in the signal compared to the rest state. The study is performed carefully and the authors provide evidence in support to their conclusion. They also perform numerical simulations for a minimal model to provide an interpretation of the observed reduced variability in terms of the dynamical evolution of the systems in resting conditions and under tasks.

The manuscript attacks a very intriguing problem of wide interest in the community. My criticism mainly derives from the apparently missing lack of a systematic control of the stimulation in the experimental protocol. More precisely, I am not surprised that the variability in the signal is reduced by the tasks. This is an expected result as the reduction of the fluctuations of the magnetization in a ferromagnet by the application of a large magnetic field. The strong interaction with the field aligns all spins in the field direction. The main conclusion by the authors would be more convincing if observed systematically by controlling the amplitude of the external stimulation. The task is not well defined in the manuscript and the duration of the application of some seconds suggests that the stimulation is quite large. Moreover, I think that applying, in a sequence, different stimulations (not otherwise specified) is also a non-rigorous procedure. Indeed, it is possible (as suggested by numerical simulations) that the application of a short in time small amplitude stimulation would not reduce the variability. It would be also more appropriate to analyse the variability by averaging the response to the same task, not a series of different ones as I understand from the manuscript. I therefore suggest to analyse data for different durations of the stimulation applied, by using the same stimulation frame not a sequence of different ones, whose effects on the evoked activity can be averaged out in the statistical procedure. Finally, drawing conclusions on data collected from two humans only is also not very rigorous.

As a minor remark, the author should specify what they mean by a negative stimulation amplitude in the numerical study and how this would compare to experimental data. Moreover, since the authors relate the reduced variability to the high activity level induced by the task, how can they justify the reduced variability by a de-activation stimulation?

Reviewer #3: In behaving animals, pairwise correlation among neurons in a local network is generally reduced during evoked activity. By contrast, at larger-scale when we study interactions across brain regions we find that correlations (functional connectivity: FC) can both increase and decrease. Authors claim that taken together these observations are incompatible. In this study authors have addressed this problem. To this end, they have analysed synchrony and variability in spiking activity recorded from primates and in fMRI signals recorded from humans. They argue that unlike previous observations, task related activity quenches correlations and variability across cortical areas (fMRI measurements). They provide an explanation of reduction in cortical variability and correlation in evoked activity in terms of attractor dynamics. The key to the explanation is neuron mass model transfer function and how the attractor is shifted by the inputs.

Overall there is huge amount of work has gone into this manuscript: from analysis of spiking activity and fMRI data to simulation of neuron mass models with one and two populations. But I think manuscript needs some more work in terms of data analysis to really establish the results based on the experimental data. The major problem with the manuscript is the choice of the model which is not suitable to address the problem. Moreover, the even though the explanation based on the results seems fine it does not apply to the neuronal networks in the brain. So we have an experimental observation at the level of inter brain region correlations not only that needs to demonstrated with greater rigour but also there is no explanation of the observations. In the following I elaborate on these.

Major:

- At the outset I do not agree that there is any incompatibility in reduction in spiking activity correlations and increase/decrease in fMRI-based FC. There is no reason why these two should be similar. Spiking activity correlation reduction is a property of the local network dynamics and FC is about inter-region correlations. Not only that these correlations are measured at different time scales but the local and inter-brain region connectivity are very different.

- I agree that here authors have analysed spike level correlations using multi-unit activity between different brain regions and found that there is a reduction in both variability and correlations. Even though authors suggest that they have used consistent statistical methods to analyse spikes and fMRI signals, the underlying signals have completely different origins and are measured at very different time and spatial scales. So again, I do not think there is any reason to believe that modulation of correlation should be similar in multi-unit and fMRI signals.

- I do not think that showing just the variance of activity across trials is a proper measure of the variability. In network models when neurons are connected it may be possible that the mean and variance of the activity are inter-related. But the real reason for measuring Fanp Factor (FF) is that we want to compare with an unbiased uncorrelated population of homogeneous Poisson type spike train ensemble. Change in the variance between resting and evoked activity state could simply be because of a change in the mean responses. So I insist on showing FF as a measure of the variability.

- An additional confound is that here authors are using multi-unit spike trains. The variance of multi-unit spike train can also change if the correlation between single-units (which constitute the multi-unit) changes (even if weakly) -- variance of multi-unit spike train is sum of the variance and co-variance of single units that constitute the multi-unit. So if different populations of neurons are active in evoked and task related state, the comparison of variance becomes very tricky.

- The difference in evoked and ongoing spike rates are very small? Why is it so?

- Authors argue that in task-state correlation increase in the replication subject is because of a decrease in unshared variance (Page 7 first para). I am not sure I understand the line of argumentation. Could authors format this in mathematical equations.

- In the section 'Task-state FC ...' authors write that 'Our current finding illustrate that mean-field spike count correlation decrease...'. Are they referring multi-unit spiking activity as mean-field activity -- if so this is wrong.

- In the same section 'Task-state FC ...' authors hypothesize that FC would be globally reduced. What is the basis of this hypothesis? A priori there is no reason to assume that just because correlations in local networks are reduced that the correlation between two different areas (or the FC) should also be reduced. As said earlier, fMRI FC and spike correlation are measured at very different temporal scales and most likely have different origin.

- In Figure 3g authors show that FC is reduced in task evoked state. But this is average FC. Figure 3H is really uninformative. I do not understand what is the purpose of showing that FC of each brain area is on average reduced. If authors want to show that FC is reduced then they need to show that FC for each pair of brain regions is decreased in evoked state as compared to the ongoing activty. So I suggest that authors show a scatter diagram with x-axis as the FC of a pair in ongoing state and y-axis as FC of the same pair in evoked state. Panel 3i is good but please do not make a binary decision and show the full histogram of pair-wise difference in FC in the two states.

- Authors write that inter-area cortical network rely on excitatory connectivity -- this is correct but this does not mean that excitatory input impinges on excitatory neurons only. Long range connections impinge on both excitatory and inhibitory neurons.

- Page 16 first para: "We found a highly negative association between mean and variance under experimental perturbation, suggesting that at the mean-field level, mean and variance cannot be mechanistically dissociated" -- This is correct when we assume that the operating point of the network is right at the middle of the Sigmoid function. Authors have conveniently chosen a range of 0-1Hz for activity. In reality a network is operating at 1-2Hz (per neuron) baseline and stimulus evoked activity can push the network to very high firing rates transiently but certainly up to 20Hz in steady state. But cortical network activity will not reach saturation range -- if it does please provide suitable references. So the input induced reduction in variability while correct in the chosen model, does not make sense for a cortical network -- biological neural networks, irrespective of the spatial scale we observe them do not work in a saturation regime during evoked activity whereas in this model evoked activity induced reduction in variability is reached when the network moves towards saturation. Authors have also ignored their own data which shows that multi-unit firing rates different by a small amount in evoked and ongoing activity states -- that is even with the chosen Signoid function input induced shift in the operating point by input will be only 0.11 Hz.

- The bidirectionally coupled network of two neuron mass models is way too simple to comment on the FC measured across many brain regions. For a start they have ignored the fact that there is a sizeable delay in inter-brain region connectivity. This model is a gross oversimplification. And even with this simplification, the mechanism of stimulus induced reduction in correlation is not becoming apparent. There is no definition of correlation or variance and neither these are related to the characteristic time scales. Even if we accept the arguments authors have made, it is important to show that this argument holds true when we consider a network of N (=no of voxels in fMRI or brain regions) neuron mass models, not just 2.

Minor:

Methods: I am confused about the spike rates. In the first para of the methods subsection Spiking data: Task vs rest variability analysis explains how mean responses were subtracted from each trial. But then in the very next para they have used expression such as 'mean firing rate'. Where is the change in the firing rate used and displayed? And I hope that in eq 1 it is the true rate not a mean subtracted rate.

- Authors wrote in the introduction "In the animal spiking literature," This is a rather sloppy expression and language. Humans are animals as well.

- Figure 2: Could authors display the results as a scatter diagram with x-axis as the firing rate in resting state and y-axis as the firing rate in task related state. This way we can see how firing rate at individual channels changed in task related state. In current display scheme it is not each to see if activity at individual channel showed significant decrease or not. Same scatter diagram can be rendered for panels c and d.

- Figure 2: The caption says that panel b shows average spike rate across all recordings but the figure actually shows Spike rate change.

- Figure 2: In panels b-d what does each single dot mean? Is it average of a single session?

- Page 12: Authors write "We hypothesized that decreases in neural variability and correlations are related to task-evoked activity." But this is not really a hypothesis. Authors must explain why they hypothesize so and what kind of relationship they expect? Same applies to several other hypotheses mentioned in the manuscript. For instance, there is a hypothesis mentioned in the last para on page 12 but that refers to information representation. But on page 13 first para authors start a sentence as 'Consistent with our hypothesis....' But this refers to dimensionality not to information representation. Dimensionality and information representation are two different concepts even if they stem from the fact that correlations are reduced. Better information representation does not predict high-dimensionality.

Also a short remark on the correlation and information representation -- authors have cited the Averbeck review for this. But please note that the correlation mentioned in the context of information representation were measured at 10s of milisec and FC of FMRI is measured at much slower time scales than that. Information representation across brain areas is not the same as across neurons in a single local network.

- Figure 7c,d: The two variables do not seem to be 'perfectly correlated'

- Authors write on Page 3 "...the mechanism underlying these reduced correlations remains unclear." This is not correct. The review by Doiron provides an exhaustive overview of various mechanisms that may underlie reduction in correlation and variance e.g. clustered connectivity (Litwin-Kumar and DOiron 2012, Deco and Higher 2012), tight and detailed balance of excitation and inhibition (Hennequin et al.) or stimulus/input (Bujan et al. 2015)

**Have all data underlying the figures and results presented in the manuscript been provided?**

Reviewer #1: Yes

Reviewer #2: None

Reviewer #3: Yes

PLOS authors have the option to publish the peer review history of their article (what does this mean?). If published, this will include your full peer review and any attached files.

Reviewer #1: No

Reviewer #2: No

Reviewer #3: No
---

## [Decision Letter · Decision Letter 1]

16 May 2020

Dear Mr. Ito,

Thank you very much for submitting your manuscript "Task-evoked activity quenches neural correlations and variability across cortical areas" for consideration at PLOS Computational Biology. As with all papers reviewed by the journal, your manuscript was reviewed by members of the editorial board and by several independent reviewers. The reviewers appreciated the attention to an important topic. Based on the reviews, we are likely to accept this manuscript for publication, providing that you modify the manuscript according to the review recommendations.

Apart from the few remaining points, that I nonehteless invite you to carefully consider and address, some major disagreement on the model and its implication remain open, and is it clear that cannot be addressed without completely change the scope and idea of the paper. While the acceptance decision will not be affected by these disagreements, we think that is useful for the community to discuss around these topics. This is thus one case in which having the reviews published together with the paper would be crucial and beneficial for open discourse.

We hope you will agree to having the reviews published for this paper.

Sincerely,

Daniele Marinazzo

Deputy Editor

PLOS Computational Biology

Daniele Marinazzo

Deputy Editor

PLOS Computational Biology

[LINK]

Reviewer's Responses to Questions

**Comments to the Authors:**

Reviewer #2: The authors have addressed my comments and changed the manuscript accordingly.

Reviewer #3: The authors have done a great job in addressing most of the concerns I raised in the previous round of reviews. The manuscript is a very good shape and has some very interesting results. There are a few minor issues that still bother me.

I strongly object to the following statement in the introduction.

"These differing perspectives of the role of neural correlations in (large-scale) human imaging and (local circuit) animal neurophysiology studies appear to be incompatible. "

As I said in my previous comments, there is no reason why they should be consistent.

Both spiking activity and fMRI are brain's activity -- they have to be related. But not in a liner manner. It is a way non-linear relationship. That is, even if first moments of the spiking activity distribution and fMRI signals are related there is no reason to believe that other moments (second in this case) will also be. So I think author should remove the sentence. I think it is perfectly fine to quantify the relationship between noise correlations and functional connectivity -- that's a good enough motivation.

Similarly, while mechanistic basis of FC in fMRI remains unclear the same is not correct for spiking activity NC and reduction in NC. A number of papers have discussed this (e.g. Tetzlaff et al. 2012 PloS Comp., Renart et al. 2010 Science, Bujan et al. 2015 JNeurosci, Ecker et al. and several of papers from Dorion/Eric Shea-Brown groups).

Next, I do not think that the difference in the temporal dynamics of the spiking activity signals and fMRI can be alleviated simply by down sampling the spiking activity signals. The difference in the dynamics also means that the mechanisms underlying the statistical properties also operate at different time scales. So comparing statistical properties of spiking activity and fMRI is a rather tricky task.

Finally, I am not convinced that the model is suitable to address the problem and even if this oversimplified model gives some insights, the question of scaling remains -- i.e. from two interacting populations to N interacting populations. It is a non-trivial task so I do not expect that the authors undertake this task for this manuscript. From my own experience, it almost never straightforward to translate insights from a two population network to N-population model.

**Have all data underlying the figures and results presented in the manuscript been provided?**

Reviewer #2: Yes

Reviewer #3: Yes

PLOS authors have the option to publish the peer review history of their article (what does this mean?). If published, this will include your full peer review and any attached files.

Reviewer #2: No

Reviewer #3: No
---

## [Editor Report · Decision Letter 2]

27 May 2020

Dear Mr. Ito,

We are pleased to inform you that your manuscript 'Task-evoked activity quenches neural correlations and variability across cortical areas' has been provisionally accepted for publication in PLOS Computational Biology.

Best regards,

Daniele Marinazzo

Deputy Editor

PLOS Computational Biology

Daniele Marinazzo

Deputy Editor

PLOS Computational Biology

---

## [Editor Report · Acceptance letter]

29 Jun 2020

PCOMPBIOL-D-20-00004R2 

Task-evoked activity quenches neural correlations and variability across cortical areas

Dear Dr Ito,

I am pleased to inform you that your manuscript has been formally accepted for publication in PLOS Computational Biology. Your manuscript is now with our production department and you will be notified of the publication date in due course.

With kind regards,

Laura Mallard
